



# Evaluating tropospheric humidity from GPS radio occultation, radiosonde, and AIRS from high-resolution time series

Therese Rieckh[1,2], Richard Anthes[1], William Randel[3], Shu-Peng Ho[1], and Ulrich Foelsche[4,2]

[1]COSMIC Program Office, University Corporation for Atmospheric Research (UCAR), Boulder, Colorado, USA
[2]Wegener Center for Climate and Global Change (WEGC), University of Graz, Graz, Austria
[3]National Center for Atmospheric Research (National Center for Atmospheric Research), Boulder, Colorado, USA
[4]Institute for Geophysics, Astrophysics, and Meteorology/Institute of Physics, University of Graz, Austria

*Correspondence to:* Therese Rieckh (rieckh@ucar.edu)

**Abstract.** While water vapor is the most important tropospheric greenhouse gas, it is also highly variable in both space and time, and water vapor concentrations range over three orders of magnitude in the troposphere. These properties challenge all observing systems to accurately measure and resolve the vertical structure and variability of tropospheric humidity. In this study we characterize the humidity measurements of various observing techniques, including four separate Global Positioning System (GPS) Radio Occultation (RO) humidity retrievals (UCAR direct, UCAR 1D-Var, WEGC 1D-Var, Jet Propulsion Laboratory (JPL) direct), radiosonde, and Atmospheric Infrared Sounder (AIRS) data. Furthermore, we evaluate how well the ERA-Interim reanalysis and National Centers for Environmental Prediction (NCEP) Global Forecast System (GFS) model perform in analyzing water vapor at different levels. To investigate detailed vertical structure, we used time–height cross sections over specific locations (radiosonde stations in the tropical and subtropical western Pacific) for the year 2007. We found that RO humidity has comparable or better accuracy than both radiosonde and AIRS humidity over 800 hPa to 400 hPa, as well as below 800 hPa if super-refraction is absent. The various RO retrievals of specific humidity agree within 20 % in the 1000 hPa to 400 hPa layer, and differences are most pronounced above 600 hPa.

## 1 Introduction

Tropospheric humidity is one of the key parameters driving weather and climate, and plays an important role in the development of many extreme events. To accurately model current and future climate, it is crucial to understand the distribution, transport, and vertical structure of tropospheric water vapor. However, measuring water vapor accurately is a great challenge, as it is highly variable on both spatial and temporal scales, and its tropospheric concentration varies over three orders of magnitude between the tropical planetary boundary layer and the tropopause. At present, no single observing system can provide accurate tropospheric humidity data on a global scale with high vertical resolution.

Passive (microwave and infrared) nadir-sounding systems provide data globally, but with low vertical resolution. Weighting functions are used to derive vertically resolved humidity information, and these vertical scales are large (2 km to 3 km) compared to the variability of water vapor in the vertical. Furthermore, infrared based systems cannot provide data within or below clouds.



Radiosonde (RS) balloon measurements are launched globally, although with sparse coverage in many areas, such as over oceans or in the southern hemisphere. They can have a high vertical resolution, but data quality varies strongly depending on the sensor type (Miloshevich et al., 2006; Ho et al., 2010). Operational weather forecasting still benefits greatly from RS measurements, but the current global RS network is neither designed nor suitable for detecting and monitoring climate change.

First, many different sensor types are used globally, each with their unique known and unknown biases. Second, sensor types at different locations change over time and these changes have been poorly documented in the past, which can lead to artificial trends or jumps in the station's record (Dai et al., 2011). The GCOS (Global Climate Observing System) Reference Upper-Air Network (GRUAN) aims to address this issue by providing long-term high-quality vertical profiles of selected essential climate variables, including an estimate of the measurement uncertainty (Bodeker et al., 2016). GRUAN will play an important role for

calibrating data from other global networks, however, at this point in time certified data are available at only a few locations with a relatively short time range (less than four years).

Research aircraft can provide high-quality, high-resolution profiles, but these missions are infrequent and cannot provide a complete global picture continuously over time by themselves. They are, however, important to evaluate measurements from other observing systems or models (Rieckh et al., 2017).

The Global Positioning System (GPS) Radio Occultation (RO) technique provides near-vertical profiles of refractivity with high vertical resolution and high accuracy and precision. Other features of the RO technique are global coverage, all-weather capability, and SI-traceability. Profiles penetrating down into the lower troposphere became available with open-loop tracking (Sokolovskiy et al., 2006). Since refractivity depends on temperature and water vapor pressure, tropospheric specific humidity can be derived from refractivity via a so-called direct retrieval (using ancillary temperature information) or a One-Dimensional

Variational retrieval (1D-Var), which finds the optimal solution for water vapor pressure, temperature, and refractivity taking their prescribed errors into account. Thus the RO water vapor retrievals and their quality vary depending on the a-priori (and the accuracy of the prescribed data) and inversion method used. Several RO processing centers currently provide RO water vapor profile retrievals: University Corporation for Atmospheric Research (UCAR), Jet Propulsion Laboratory (JPL), Danish Meteorological Institute (DMI), and Wegener Center for Climate and Global Change (WEGC).

The above observing techniques have been used to investigate the global humidity distribution, trends, and radiative impact. RO, despite being a relatively young observing technique, has shown the potential to provide data of climate benchmark quality for refractivity and temperature between about 8 km and 25 km (Ho et al., 2009, 2012; Steiner et al., 2013). The quality of RO humidity is subject of research since ancillary data are required to retrieve humidity from refractivity. Kursinski et al. (1995) provided a first estimate for water vapor accuracy of less than 5 % for individual profiles in the boundary layer, and 20 % up

to about 7 km. Chou et al. (2009) found humidity differences smaller than 40 % below 7 km for individual profile comparisons between dropsondes and RO. For observations near strong typhoons, they found differences up to 100 % in the mid and upper troposphere. Regarding global specific humidity distributions, Chou et al. (2009) found good agreement within 15 % between RO and Atmospheric Infrared Sounder (AIRS), but significant discrepancy between NCEP/NCAR reanalysis and RO humidity. Ho et al. (2010) showed that UCAR COSMIC (Constellation Observing System for Meteorology, Ionosphere, and Climate)

water vapor profiles agree well with those of European Center for Medium-range Weather Forecasts (ECMWF) analysis over



different regions, demonstrating the quality of the RO humidity data. Furthermore, they used RS and RO co-located data to identify biases of various RS types. Wang et al. (2013) also used UCAR COSMIC water vapor products and global RS data with very strict co-location criteria (1 h, 100 km) to verify the quality of UCAR RO humidity and found a mean specific humidity bias of $-0.012\,\mathrm{g\,kg^{-1}}$, with a standard deviation of $0.666\,\mathrm{g\,kg^{-1}}$ over the 925 hPa to 200 hPa layer. Ladstädter et al. (2015)

compared WEGC RO profiles from multiple missions to a five year record of GRUAN RS profiles (both of which have the potential to serve as reference observations in the GCOS) and to a standard 11 year record of RS profiles (Vaisala RS90/92). Vaisala RS90/92 shows a dry bias of 40 % in the troposphere compared to RO, whereas GRUAN, with an elaborate humidity bias correction scheme, agrees within 5 % with RO below 300 hPa. Ladstädter et al. (2015) state that the good agreement of the RO and GRUAN RS data sets strongly encourages further development and advancement of both systems for the benefit

of future climate monitoring and research. Vergados et al. (2015) compared relative humidity of JPL RO, ECMWF Reanalysis Interim (ERA-Interim), and Modern-Era Retrospective analysis for Research and Applications (MERRA) in the tropics and showed that from a climatological standpoint, MERRA and JPL RO are in agreement, whereas the ECMWF reanalysis is drier. Vergados et al. (2017) compared JPL and UCAR RO humidity data sets to MERRA, ERA-Interim, and AIRS from 2007 to 2015 for the $\pm40°$ latitude range between 700 hPa and 400 hPa. They found that the both RO humidity retrievals agree well

with MERRA and ERA-Interim, but the JPL retrieval is overall moister than all other data sets, while both the UCAR retrieval and AIRS are overall drier than all other data sets.

All of the above work considered differences averaged over large geographical regions and long time periods (a month or longer). While useful for climate and error estimations, these averages obscure variability that takes place on smaller temporal and spatial scales. Case studies fill this gap, but they often focus on a single, particular event that occurs over only a few days.

The objectives of this study are to i) quantify RO humidity retrievals in the troposphere using four RO datasets where different retrieval methods and ancillary data were used, and ii) quantify how these RO humidity data sets differ from AIRS, RS humidity measurements, and the ERA-Interim and GFS model analyses.

We zoom into water vapor variability in both a temporal and spatial sense by comparing data from multiple observing techniques (RO, RS, AIRS) and model (re)-analyses (ERA-Interim, Global Forecast System (GFS)) at particular locations in

the tropics and sub-tropics over an entire year. We chose the year 2007, when the maximum number of COSMIC RO profiles was available (COSMIC was launched in 2006 (Anthes et al., 2008)). We compare each of these individual data sets with co-located ERA-Interim humidity results for a) the surface to the upper troposphere, b) four locations, c) four seasons, and d) during typhoon passages. We quantify the structural uncertainty of RO derived humidity profiles in the troposphere, which results from different inversion implementations and a-priori. To understand how the RO humidity data sets are different from

other humidity products, we collected RS–ERA pairs, AIRS–ERA pairs, and GFS–ERA pairs near the four RS stations. These data pairs may not sample the same local times, but errors due to local time sampling difference are assumed to be small since water vapor errors over these locations may not vary significantly at different local times.

As humidity varies strongly in time and space, this study allows us to show in detail how humidity conditions change over time, both daily and seasonally, and how atmospheric conditions affect the ability of these data sets to provide accurate



and precise humidity information. We can identify high-frequency variability and patterns at selected locations that would be obscured if only statistical parameters were analyzed.

We focus on several challenging locations in the tropics and sub-tropics where water vapor is highly variable. We show the entire 1000 hPa to 400 hPa range to show how data quality for different observing and modeling systems varies with altitude.

For example, the humidity data from many RS sensors are biased in the mid and upper troposphere. RO-derived humidity can be biased in the lowest few kilometers (due to super-refraction in the atmosphere) and is unreliable once temperatures get as low as 250 K in the upper troposphere (around 350 hPa in the tropics). Using data from 1000 hPa to 400 hPa without layer averaging allows us to identify details in the vertical humidity structure as measured by these systems.

ERA-Interim Reanalysis (hereafter ERA) is used as reference for all comparisons. Although all data sets used in this com-

parison are assimilated in the ERA, comparisons are still valuable since i) data from a large number of different observing techniques are assimilated (number of assimilated observations more than $10^7$ per day in 2010 (Dee et al., 2011), thus lowering the impact of any single observation), and ii) the RO uncertainties data assimilation are very large in the mid and lower troposphere, and hence RO makes a relatively small contribution in the ERA reanalysis. In the ERA, the standard deviation of the RO observation error distribution (in bending angle space) is assumed to decrease linearly with increasing height, from 20 %

at the surface to 1 % at 10 km impact height (Poli et al., 2010).

In a companion paper (Anthes and Rieckh, 2017), these data sets are compared statistically in different ways to estimate the error covariances of each data set. This method indicates that the ERA-Interim data set has the smallest errors in refractivity, temperature, specific humidity, and relative humidity from 1000 hPa to 200 hPa. The current paper sets the stage for this statistical comparison by describing the data sets in detail and showing how they vary over the year at the four locations.

The structure of this paper is as follows: Section 2 explains the data sets used in this study. Section 3 shows on overview of the results for the different observing systems, which are analyzed in greater detail in section 4. Section 7 provides a summary and conclusions.

## 2 Data and Method

### 2.1 Radio occultation

Radio occultation (RO) is a limb sounding technique that provides near-vertical profiles with high vertical resolution of bending angles (Melbourne et al., 1994; Hajj et al., 2002), which can be used to retrieve atmospheric refractivity $N$. $N$ can be related to atmospheric temperature $T$, pressure $p$, and water vapor pressure $e$ via the Smith–Weintraub formula (Smith and Weintraub, 1953):

$$N = 77.6 \frac{p}{T} + 3.73 \times 10^5 \frac{e}{T^2} + [...] \tag{1}$$

The contribution to $N$ from liquid water (the terms in [...] in Eq. (1)) can be neglected in most conditions (Ho et al., 2017). When $e$ is negligible (at temperatures lower than 250 K (Kursinski et al., 1997)), the second term is assumed zero and atmospheric temperature can be computed using Eq. (1).



In the troposphere, where water vapor content is significant, Eq. (1) is ambiguous and ancillary temperature data from another data source (usually model or analysis temperature) are required to solve for $e$. Direct retrievals use a prescribed $T$ from another source to derive $e$. In a One-Dimensional Variational (1D-Var) retrieval, a cost function is minimized to find the optimal solution for $e$, $T$, and $N$ with their prescribed errors (Poli et al., 2002). In this study, three different RO retrievals and

four different humidity retrievals are compared in order to provide an indication of the uncertainty in RO-derived water vapor.

### 2.1.1 UCAR 1D-Var

A One-Dimensional Variational (1D-Var) retrieval generally uses an a-priori state of the atmosphere (background vertical profile), an observable (RO refractivity or bending angle), and their specified associated errors to minimize a quadratic cost function. At COSMIC Data Analysis and Archive Center (CDAAC), ERA profiles of temperature and humidity are used as

background, which are interpolated to the time and location of the RO (accounting for tangent point drift during the occultation). The humidity retrieval allows an error for both $T$ and $e$, but only a very small error for bending angle/refractivity. CDAAC provides the resulting profiles of $N$, $T$, $e$, and $p$ (wetPrf[1]), hereafter called UCAR 1D-Var.

### 2.1.2 UCAR direct

A direct retrieval uses background temperature and RO refractivity to compute humidity using Eq. (1). The influence of a $T$

error on $e$ (i.e. the relation between $\delta T$ and $\delta e$) can be directly derived from Eq. (1) (Ware et al., 1996), under the assumption, that $N$ and $p$ are constant:

$$dN = \frac{\delta N}{\delta T}\delta T + \frac{\delta N}{\delta e}\delta e = 0 \longrightarrow \delta e = \frac{1}{3.73 \times 10^5}(2NT - 77.6p)\delta T \qquad (2)$$

Ware et al. (1996) showed that $e$ could be estimated to within 0.25 hPa in the lower troposphere if temperature were known to within 1 K. Vergados et al. (2014) depict the specific humidity retrieval errors due to temperature uncertainty for several

latitude bands and pressure levels and show that humidity errors increase with increasing altitude and latitude, since humidity decreases and thus its contribution to atmospheric refractivity. In the tropics (relevant for this study), the $q$ uncertainty for 1 K $T$ uncertainty is less than $\pm 3\,\%$ below 700 hPa and increases to 18 % at 400 hPa (cut-off altitude in this study).

We use the RO variable "N_obs" (observed $N$ before going through the 1D-Var) from the UCAR CDAAC wetPrfs. We chose $T$ from the co-located GFS profiles as prescribed temperatures in the humidity retrieval for a greater independence between

RO and ERA. For the four locations in this study, the maximum $T$ difference between GFS and ERA occurs at Guam, with up to 2 K in the 800 hPa to 500 hPa layer for the individual profiles. Comparisons of the UCAR direct retrieval using GFS $T$ versus ERA $T$ as background temperature shows specific humidity differences of less than 3.5 % for seasonal and 200 hPa layer averages within the 800 hPa to 300 hPa layer.

---

[1]http://cdaac-www.cosmic.ucar.edu/cdaac/



### 2.1.3 WEGC 1D-Var

The Wegener Center for Climate and Global Change (WEGC) developed a simplified version of a 1D-Var method. As a background, they use ECMWF 24 h or 30 h forecast fields, which are spatially interpolated to the location of the RO (Schwärz et al., 2016). Combining the Smith–Weintraub equation and the hydrostatic equations for dry and moist air, they are solved for $e$ and $p$ with prescribed $T$, and for $T$ and $p$ with prescribed $e$. Iteration continues until the retrieved $e$ and $T$ converge within a set tolerance. Then the results are combined to get the optimally estimated $T$ and $e$ profiles. More information about the retrieval and error characteristics can be found in Ladstädter et al. (2015) and references within.

### 2.1.4 JPL direct

JPL's direct retrieval is similar to the UCAR direct, but uses the ECMWF Tropical Ocean and Global Atmosphere (TOGA) $T$ as a-priori. Humidity is only derived below the level of tropospheric $T = 250$ K (Kursinski et al., 1997). JPL RO data were downloaded via the Atmospheric Grid Analysis and Profile Extraction tool[2].

## 2.2 ERA-Interim Reanalyses

We use the ERA as a reference for our comparisons[3]. We consider the ERA to be the most accurate data set (Anthes and Rieckh, 2017) because it uses quality-checked observations with a 4D-Var data assimilation scheme and an accurate forecast model in a research mode to produce the variables of interest here (temperature and water vapor) on a $0.7° \times 0.7°$ grid. In 2007 ERA assimilates measurements from many different observing techniques, including RS observations, AIRS radiances, and RO bending angles (Dee et al., 2011). Apart from using ERA as reference, we also created two baseline data sets from ERA for comparison to the observations. The first one is climatology (hereafter CLIMO) for 2007, which is simply the ERA 2007 annual mean. The second one is the persistence (PERSIST) value of each variable from the value of the time series 24 hours earlier. It represents a measure of the day-to-day variability in the ERA data set. These two simple data sets represent a baseline against which the value of observations can be compared. A minimum requirement for an observation type to be useful is that it must contribute additional information above those contributed by these baseline data sets, i.e. they must be more accurate than these data sets.

## 2.3 Radiosonde, AIRS, and GFS

RS data for Guam (13.5°N, 144.8°E) and 3 Japanese stations (Ishigakijima: 24.2°N, 124.5°E; Minamidaitojima: 25.6°N, 131.5°E; Naze: 28.4°N, 129.4°E) (Fig. S1, supplement) were downloaded from the National Oceanic and Atmospheric Administration[4]. The RS are given on six main pressure levels between 1000 hPa and 400 hPa, plus additional levels if there is higher resolution vertical structure. The RS at the four stations are generally launched twice daily during the hour before

---

[2]https://genesis.jpl.nasa.gov/agape/

[3]https://rda.ucar.edu/datasets/ds627.0/

[4]https://www.ncdc.noaa.gov/data-access/weather-balloon/integrated-global-radiosonde-archive





midnight and noon, UTC. The four stations use the following sensors: Guam: VIZ/Sippican B2; Ishigakijima: Meisei; Minamidaitojima: Vaisala RS92; and Naze: Meisei[5]. The VIZ/Sippican B2 humidity sensor has a nighttime wet bias (Wang and Zhang, 2008; Ho et al., 2010), and performs poorly in dry conditions (H. Vömel, personal communication, 2017). Ho et al. (2010) found no obvious bias for the Meisei sensor. The Vaisala RS92 sensor is known for its dry bias (Vömel et al., 2007) of $\sim$9 % at surface, and up to 50 % at 15 km altitude, and several correction schemes have been developed to address this (Miloshevich et al., 2006; Vömel et al., 2007).

AIRS is a nadir looking instrument aboard the National Aeronautics and Space Administration (NASA) Aqua satellite, which was launched in May 2002. AIRS provides atmospheric variables on 28 standard pressure levels between 1100 hPa and 0.1 hPa (8 levels between 1100 hPa and 400 hPa)[6]. The vertical resolution is $\sim$1 km for temperature and $\sim$2 km for humidity[7]. The horizontal resolution[8] is 50 km. We use the AIRS Version 6 Level 2 (AIRS2RET) data with a quality flag of BEST or GOOD.

RO co-located profiles for GFS are added in the comparison to show results from an analysis that is different from ERA. GFS profiles are given on a 25 hPa or 50 hPa grid (depending on altitude) and are linearly interpolated to the time and location of the UCAR 1D-Var profiles.

## 2.4 Design of the comparisons

Since we are investigating humidity differences of various observing systems, we chose regions where humidity conditions are highly variable in both space and time with extremely high and low values during the year. We use the tropical location Guam, which frequently experiences dry air intrusions from the subtropical upper troposphere – lower stratosphere (UTLS) region from December to March (Rieckh et al., 2017). This leads to sharp vertical humidity gradients (relative humidity changes from less than 10 % to about 80 % within a small vertical layer), conditions, that are favorable for RO super-refraction (Garratt, 1992). Super-refraction, in turn, will lead to a negative bias in the RO observed $N$ and $q$. See Fig. S3 (supplement) for the ERA 2007 time series of specific humidity, relative humidity, temperature, and refractivity at Guam.

The other RS locations are subtropical stations around Japan, which experience a large seasonal variability as well as extreme conditions associated with occasional typhoons. See Fig. S4 (supplement) for the ERA 2007 time series of specific humidity, relative humidity, temperature, and refractivity at Ishigakijima.

To increase the number of co-located profiles, we picked the year 2007 for our analysis when all COSMIC satellites were operating reliably. Since the measurement techniques for RO, RS, and AIRS are different, we use different co-location criteria to get a maximum number of high quality co-locations. For the ERA reference grid points matched to the RS stations, the distance between any of the RS stations and the respective ERA grid point is between 15 km and 35 km, and the time difference less than an hour from the 00 and 12 UTC ERA data. RO observations are co-located within 3 h and 300 km, and a co-location

---

[5]https://www1.ncdc.noaa.gov/pub/data/igra/history/igra2-metadata.txt

[6]ftp://airsl2.gesdisc.eosdis.nasa.gov/ftp/data/s4pa/Aqua_AIRS_Level2/AIRS2RET.006/

[7]http://airs.jpl.nasa.gov/data/physical_retrievals

[8]http://disc.gsfc.nasa.gov/uui/datasets/AIRS2RET_V006/summary





correction as described by Gilpin et al. (2018) is applied:

$$\Delta X_{\mathrm{SC}} = (X_{\mathrm{RO}} - X_{\mathrm{RS}})_{\mathrm{SC}} = (X_{\mathrm{RO}} - X_{\mathrm{RS}}) - (X_{\mathrm{ERA}}^{\mathrm{RO\,location}} - X_{\mathrm{ERA}}^{\mathrm{RS\,location}}) \tag{3}$$

where $\Delta X_{\mathrm{SC}}$ denotes the spatial-corrected difference of $X$, $X$ is a variable measured by RO and RS, and the co-location correction is the difference in the ERA values of $X$ at the RS and RO locations. Gilpin et al. (2018) show that this correction

significantly reduces the mean and RMS differences of the RO and RS observations. Since our reference location is an ERA grid point, we replace RS by ERA in Eq. (3), which simplifies to $\Delta X_{\mathrm{SC}} = (X_{\mathrm{RO}} - (X_{\mathrm{ERA}}^{\mathrm{RO\,location}}))$.

AIRS profiles are extracted within 30 km from the ERA reference point, the maximum time difference is 3 h. Figure S2 in the supplement depicts the co-location process for all data sets and one time stamp.

Due to the restrictions as explained above, the resulting profile pairs (and number of profile pairs) between ERA and any of

the data sets are different. Furthermore, the four RO retrievals have different quality control schemes, which especially lowers the number of available JPL profiles. The penetration depths also vary for the RO data sets and retrievals, e.g., the UCAR 1D-Var data is available on lower levels than UCAR direct because the bottom height is given as $z_{\mathrm{OB}} - e_{\mathrm{CL}}$, where $z_{\mathrm{OB}}$ is the bottom height of observation, and $e_{\mathrm{CL}}$ is the background error correlation length (which is 500 m in the UCAR 1D-Var).

All data sets are interpolated to a common 25 hPa grid. We chose this grid as a compromise between the effective resolutions

of all data sets used. The effective resolution of RO is estimated to be higher than 100 m in the troposphere (Gorbunov et al., 2004). The RS has observations on additional levels (significant levels) if there are significant changes in the vertical profile. ERA and GFS are provided on a pressure grid with 25 hPa or 50 hPa increments. AIRS is sampled on a sparser vertical grid, and thus does not resolve small scale features in the vertical. But any biases over deep layers will be evident, and if interpolation leads to biases in certain pressure layers, a pattern will be clearly visible in the individual profiles.

Profile pairs of ERA and each data set are extracted, and the computed differences are normalized by the ERA 2007 mean value (CLIMO) at each level: ND = (data set − ERA)/CLIMO. To make it easier to transfer results from normalized to actual differences, the constant value CLIMO is used to normalize all data sets. The values for CLIMO are shown in Fig. 1 and the exact values are provided in the supplement in Table S1 for an easy reproduction of the original values.

## 3   Results

### 25   3.1   Overview: General agreement and correlation between the data sets

Figure 2 shows values of $q$ for UCAR direct, UCAR 1D-Var, WEGC 1D-Var, JPL direct, RS, AIRS, and GFS (left to right) versus ERA from high to low pressure layers (top to bottom), depicting the correlation between the observational data sets and ERA at Guam. There is good agreement and high correlation for all data sets in the 1000 hPa to 400 hPa layer (Fig. 2, bottom panels). The RS shows the largest difference (∼ one order of magnitude) for generally low humidity values. Some

larger differences can also be seen for the UCAR direct, UCAR 1D-Var, JPL direct, and GFS, when these data sets are much drier than ERA (primarily happening in the DJF season). The data sets look similar in the 400 hPa to 300 hPa layer, and a dry bias for the RS becomes visible. In the 300 hPa to 200 hPa layer, the UCAR direct spread becomes very large (indicating



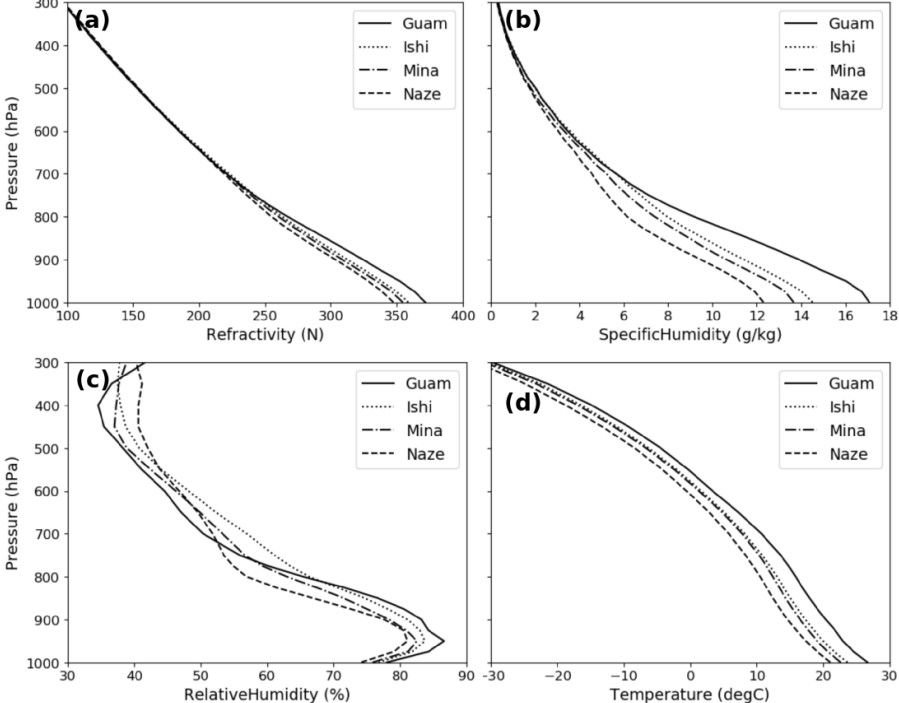

**Figure 1.** ERA annual average profiles on the 25 hPa grid for (a) refractivity, (b) specific humidity, (c) relative humidity, and (d) temperature at all four locations.

limited usefulness for RO direct retrievals at this level), while the UCAR 1D-Var and WEGC 1D-VAR agree very well with ERA (using ERA and ECMWF short-range forecast profiles as background in the retrieval, respectively). JPL direct humidity data are not available at these pressure levels. Both RS and AIRS show a dry bias. Finally, in the 200 hPa to 100 hPa layer the UCAR direct data are useless, the UCAR 1D-Var is practically identical with ERA (simply recovering ERA a-priori values), and the RS and AIRS data both have a strong dry bias. The GFS agrees fairly well with ERA in the upper layers and has no obvious bias.

### 3.2 Timeseries at Guam

Figure 3 (a) shows the time-height cross section of relative humidity (RH) over 2007 from 1000 hPa to 400 hPa at Guam. Overall, the conditions at Guam are moist (RH>80 % and $q \sim 17 \, \text{g kg}^{-1}$) year-round in the boundary layer and in the mid troposphere from July to November, and dry in the mid troposphere during the rest of the year. The changing humidity pattern above 800 hPa results from the alternation of the tropical conditions (high humidity, strong convection) and dry air intrusions from the subtropical UTLS in December to June (Randel et al., 2016). These dry intrusions (relative humidity as low as a few percent) are very stable and suppress convection. The sharp humidity gradient between the very dry lower mid troposphere and



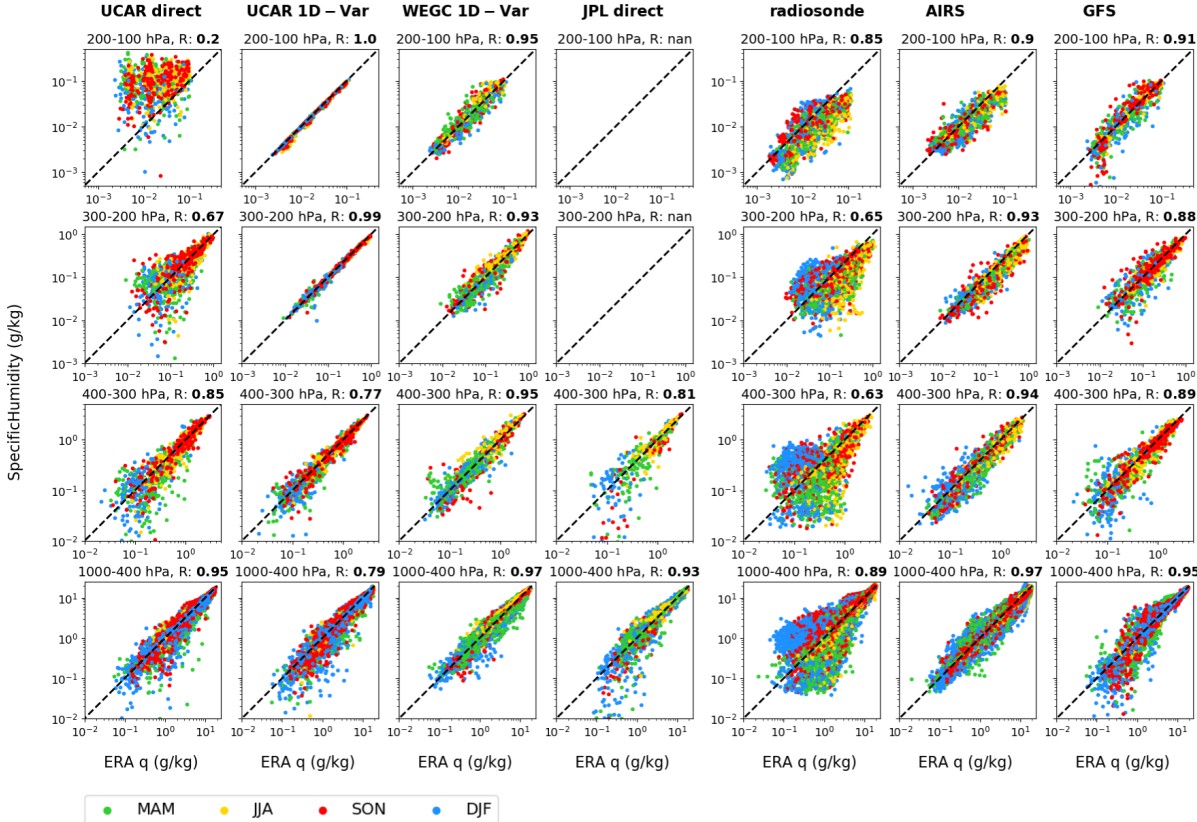

**Figure 2.** Guam: Scatter plots of normalized $q$ for 7 data sets versus ERA for 4 pressure layers. Left to right: UCAR direct, UCAR 1D-Var, WEGC 1D-Var, JPL direct, RS, and AIRS. Note that the axes are on a logarithmic scale, and that axis limits vary for different pressure layers.

the moist boundary layer around 800 hPa often leads to conditions of super-refraction, which results in a negative bias of $N$ and thus $q$ in the RO profiles (Xie et al., 2010).

The normalized difference (ND) of specific humidity $q$ between the PERSIST data set and ERA (which represents the day-to-day variability of ERA) shows that $q$ has almost no day-to-day variability in the 1000 hPa to 800 hPa layer during the entire

5    year, and in the 800 hPa to 600 hPa layer in August and September (Fig. 3 (b)). Above, day-to-day variability is significant. Exceptions occur in the 600 hPa to 400 hPa layer during December through May, when persistent dry air intrusions occur. This shows just how stable and persistent these layers can be, suppressing major changes in humidity for up to 20 days in a row.

The ND of $q$ between GFS and ERA (Fig. 3 (c)) shows that the differences between the two model values of $q$ are much smaller than the differences between PERSIST and ERA, as might be expected. GFS is up to 50 % moister than ERA in the

10    800 hPa to 600 hPa layer in the dry season, and in the 800 hPa to 550 hPa layer in the wet season. This is essentially the layer of strong humidity variability above the bottom layer of constant (about 80 %) relative humidity. This behavior may be due to GFS difficulties in capturing the sharp transition between dry and wet conditions on the bottom of dry layers in December to



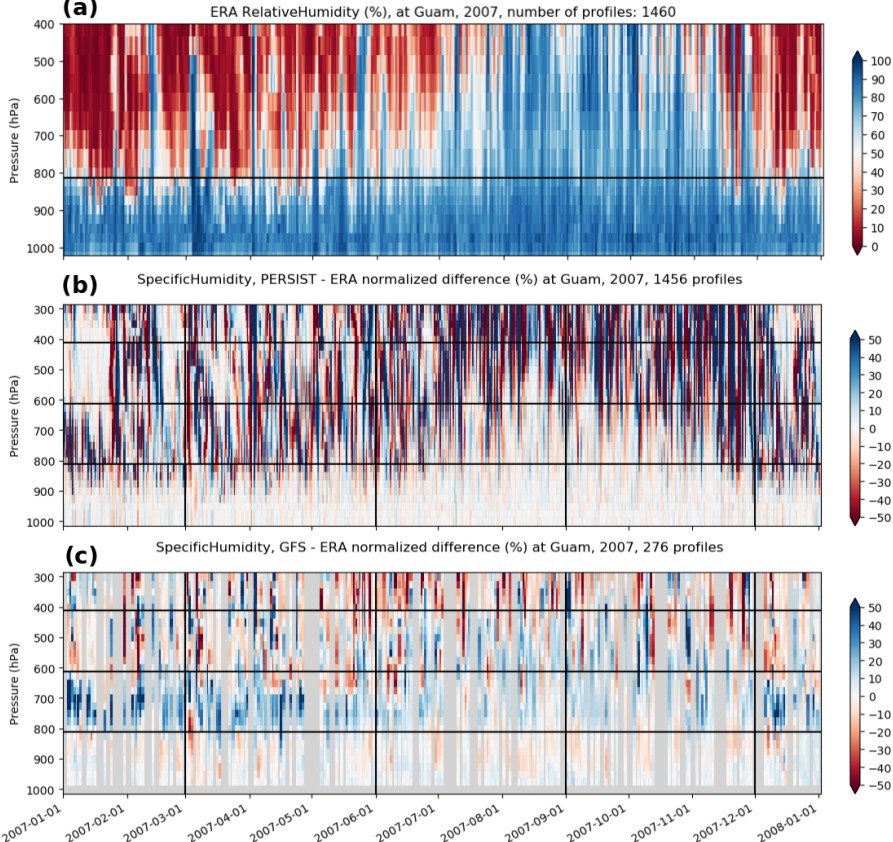

**Figure 3.** 2007 time series at Guam: (a) ERA relative humidity (%) with blue representing moist air and red representing dry air; (b) normalized difference of $q$ between PERSIST and ERA; (c) normalized difference of $q$ between GFS and ERA. The bottom panel shows that there are significant ($\pm 50\%$) differences in the two model data sets.

June. This is supported by individual profiles (e.g. Randel et al. (2016), Fig. 4), as well as our comparison of ERA with RS (Fig. 4 (a)), which supports the ERA in this respect.

The ND between RS and ERA show a small wet bias in the RS in the lower troposphere and large wet and dry biases in the middle and upper troposphere throughout the year (Fig. 4 (a)). The large biases are likely caused by RS sensor malfunctions

5  (H. Vömel, personal communication, 2017), which can start as low as at 800 hPa. At some point during the ascent, the sensor gets stuck and keeps reporting the same humidity value, which manifests itself as a dry or wet bias compared to ERA, depending on if tropospheric conditions are drier (December through May) or wetter (June through November) than the incorrect reported value.

AIRS shows an overall dry bias compared to ERA throughout the entire troposphere in all seasons (Fig. 4 (b)). The dry bias

10  appears to be less during the dry air intrusion events in the 600 hPa to 400 hPa layer in the dry season December to June. This indicates that AIRS is less biased if the overall atmospheric conditions are dry. The AIRS dry bias agrees well with the findings



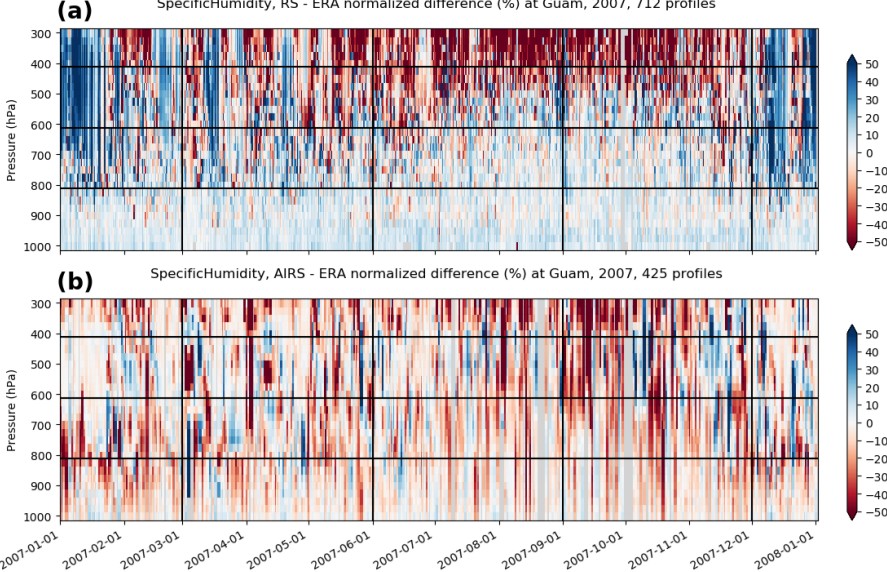

**Figure 4.** 2007 time series of the $q$ normalized difference between (a) RS and ERA, and between (b) AIRS and ERA at Guam.

of Wong et al. (2015), who studied the uncertainties of AIRS Level 2 version 6 $q$ and $T$ depending on cloud types. They found reduced dry biases in the middle troposphere under thin clouds, but a dry bias up to 30 % throughout the troposphere in the presence of thick clouds.

The normalized differences of the four RO retrievals to ERA show similar patterns in the 1000 hPa to 800 hPa layer, but larger differences in the mid and upper troposphere (Fig. 5). The UCAR direct data develop a wet bias above 600 hPa in the wet season, and alternate between dry and wet during the other seasons. The UCAR 1D-Var data show an overall dry bias throughout the troposphere with a few exceptions. Both JPL and WEGC data develop a strong wet bias above 600 hPa in the wet season. Common features of all four RO retrievals include the very small differences to ERA in the wet season in the 1000 hPa to 800 hPa layer, and a dry bias and/or frequent reduced penetration depth (loss of signal) in the dry season. The latter is a signature of super-refraction, which itself is caused by strong humidity gradients, usually between the planetary boundary layer and the free troposphere.

Figure 5 also shows a dry bias with respect to ERA in December through February in the 800 hPa to 600 hPa layer, which is clearly above the layer of strong humidity gradients (compare to Fig. 3 (a)). We conclude that this is not a dry bias in RO, but is likely a wet bias in ERA in the layer just above the strong humidity transition from wet (PBL) to dry (above). The assumed errors for assimilating RO in ERA are large in the lower troposphere, and all assimilated nadir viewing instruments only provide vertical resolutions of about 2 km to 3 km. Unless a nearby, approved RS contributes information locally, ERA does not have any vertically well resolved humidity data that will cause the ERA analysis to develop such sharp humidity gradients. This conclusion is supported by e.g. the lower right panel of Fig. 2 in Rieckh et al. (2017), where ERA data are given on the 775, 750, 700, and 650 hPa pressure levels (about 2.3, 2.6, 3.1, and 3.8 km). The 775 hPa and 650 hPa levels agree



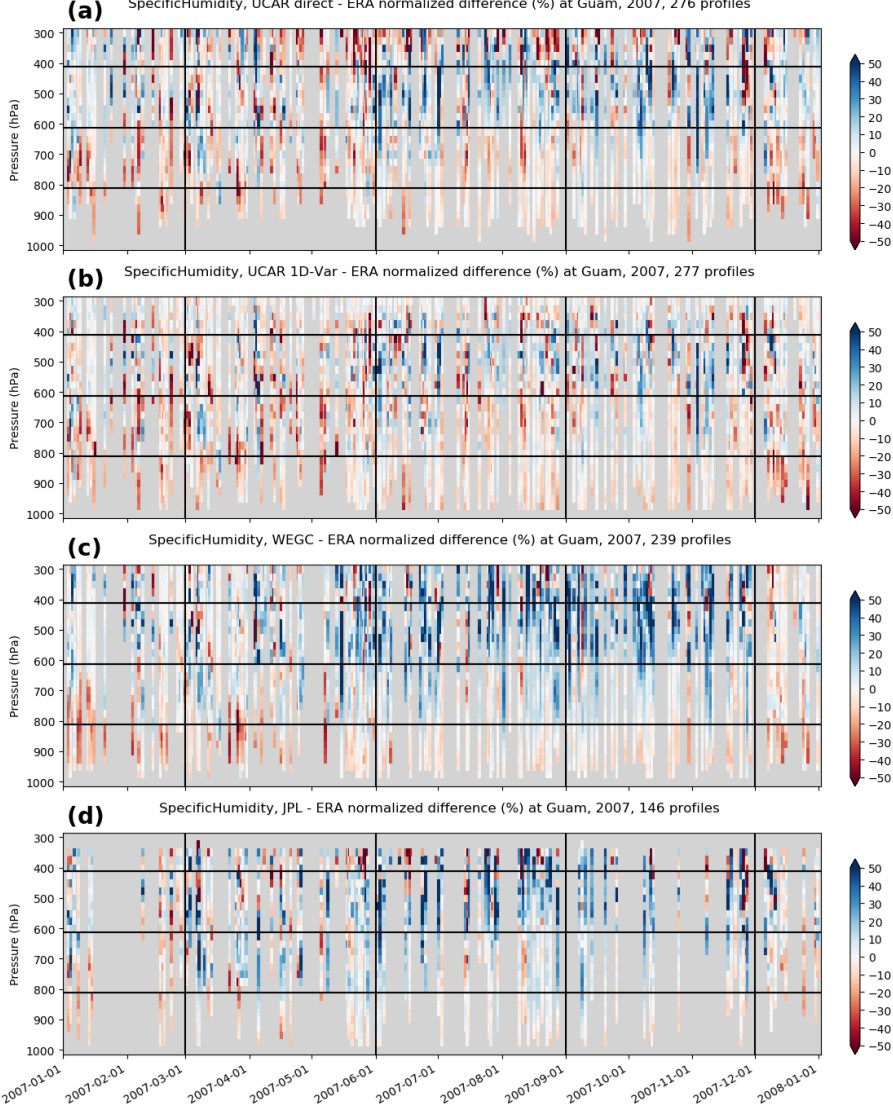

**Figure 5.** The 2007 time series of the $q$ normalized difference at Guam for the (a) UCAR direct, (b) UCAR 1D-Var, (c) WEGC 1D-Var, and (d) JPL direct retrieval.

well with the aircraft and RO measurements; however, the two levels in between smear the sharp profile and the ERA shows humidity values $1.5\,\mathrm{g\,kg^{-1}}$ to $2.5\,\mathrm{g\,kg^{-1}}$ (20 % to 35 %) larger than the observations.

The 2007 time series of the $q$ normalized difference for all data sets are depicted for a Japanese station (Minamidaitojima) in Fig. S5 and Fig. S6 (supplement).

5   The 2007 time series of the refractivity $N$ normalized difference for all data sets are depicted for Guam in Fig. S7 and Fig. S8 (supplement).





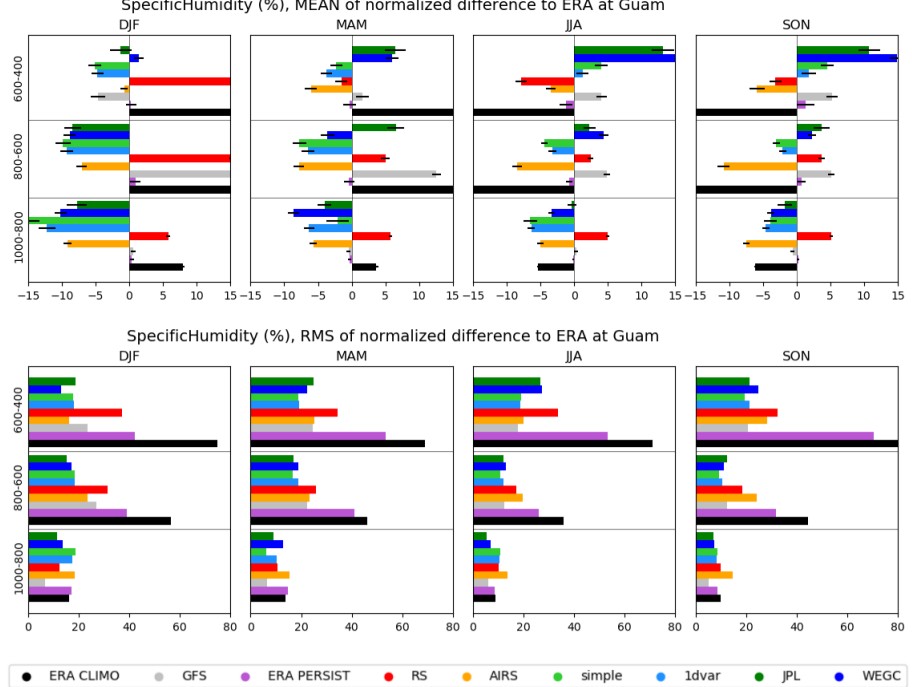

**Figure 6.** Guam: The mean (top) and RMS (bottom) of the $q$ normalized difference for all data sets, 3 pressure layers, and 4 seasons. Data sets from top to bottom (per pressure layer): JPL direct, WEGC 1D-Var, UCAR direct, UCAR 1D-Var, RS, AIRS, GFS, PERSIST, CLIMO.

### 3.3 Mean and RMS differences from ERA at Guam

We compute the mean and root mean square (RMS) of the normalized differences at Guam to get an overview of the biases and the overall differences from the ERA for all data sets for three pressure layers (1000 hPa to 400 hPa in 200 hPa layers) and four seasons (Fig. 6).

5     Some general aspects of the different data sets seen in the individual time series are clearly visible in the mean (Fig. 6, top), such as the large negative (dry) difference of RO compared to ERA (green and blue bars) in DJF for the 1000 hPa to 600 hPa layer. In the 1000 hPa to 800 hPa layer, a dry bias for RO exists throughout the year. The dry bias is largest in DJF, but it is smaller and comparable in magnitude to the biases of the RS and AIRS in MAM, JJA, and SON. RO retrievals show the greatest differences from each other in the 600 hPa to 400 hPa layer year round, and in the 800 hPa to 600 hPa layer in the wet

10   season. AIRS shows an overall dry bias at all pressure layers and seasons. As expected, PERSIST has essentially no bias at any pressure layer or season. Because of the large seasonal variation in water vapor, CLIMO has large seasonal positive and negative biases above 800 hPa that are much larger than the biases of any other data set. GFS shows significant differences from ERA, especially in the dry season (DJF and MAM) in the 800 hPa to 600 hPa layer.

    All data sets have a comparable (below 800 hPa) or considerably smaller (above 800 hPa) RMS than both CLIMO and

15   PERSIST in all seasons (Fig. 6, bottom). The former is expected, considering how little humidity changes throughout the year





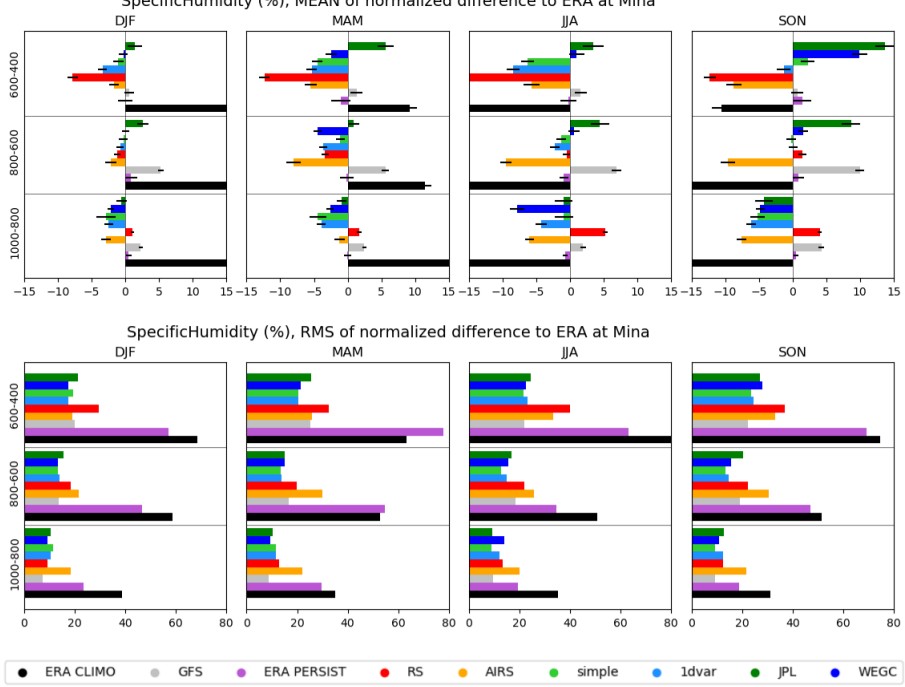

**Figure 7.** Mina: Mean (top) and RMS (bottom) of the $q$ normalized difference for all data sets, three pressure layers, and four seasons

in the 1000 hPa to 800 hPa layer. The latter indicates the value (over persistence and climatology) of all observation techniques above 800 hPa. As for the individual data sets, we see that the RO RMS for all retrievals is comparable or lower than RS and AIRS RMS for all seasons and pressure layers. This increases our confidence regarding the value of RO mid and lower tropospheric humidity data.

## 3.4 Statistics at the subtropical Minamidaitojima

At Minamidaitojima all data sets have a smaller bias compared to ERA (Fig. 7, top) than at Guam. The strong RO humidity bias in the dry season lower troposphere (as seen at Guam) is not present and biases of all observational data sets (with the exception of AIRS) are less than 5 % in the 800 hPa to 600 hPa layer. Biases are larger in the 600 hPa to 400 hPa layer, especially for the RS. The RMS values at Minamidaitojima (Fig. 7, bottom) shows a similar pattern to the one at Guam, with the RO and GFS RMS differences being smaller than the RS and AIRS differences. The statistics of the other two Japanese stations (Ishigakijima and Naze) are similar (not shown).





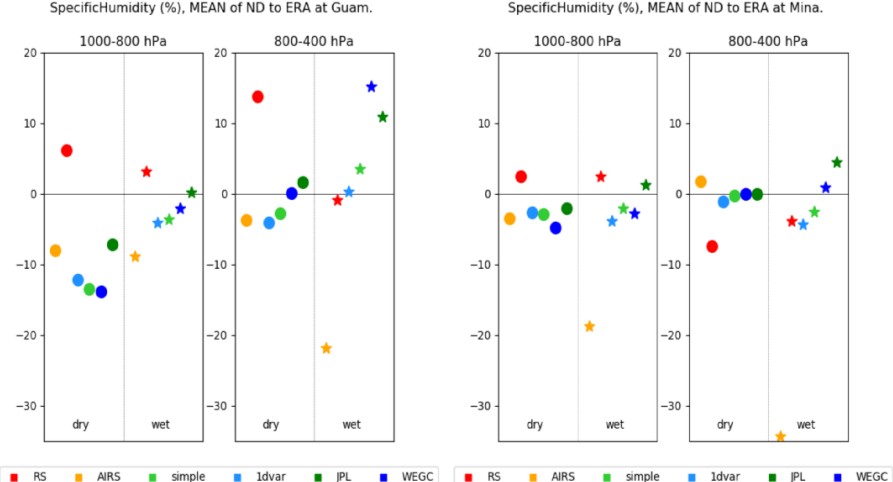

**Figure 8.** Mean differences for dry versus wet atmospheric conditions based on $\overline{\text{RH}}_{800-400}$ at Guam (left) and Minamidaitojima (right). The different colors represent the different data sets. Circles and stars represent the data sets for dry and wet conditions in the mid troposphere, respectively.

## 4 Biases in dry versus wet atmospheric conditions

In section 3 we saw how the general atmospheric humidity conditions (wet versus dry) can have an influence on the biases in the data sets, especially for RO (super-refraction with strong vertical humidity gradients) and AIRS (smaller bias in dry conditions). In this section, we investigate the different error characteristics for dry and wet conditions in more detail at both the tropical and subtropical locations. We created a "dry" and "wet" data set. For every profile pair, we computed the average relative humidity (RH) of the ERA (background) profile for the 800 hPa to 400 hPa layer ($\overline{\text{RH}}_{800-400}$). This layer was chosen according to the humidity distribution throughout the year (see Fig. 3, top). If $\overline{\text{RH}}_{800-400} \leq 30\,\%$, the entire profile is added to the "dry" data set. If $\overline{\text{RH}}_{800-400} \geq 70\,\%$, the entire profile is added to the "wet" data set. Then the mean and RMS are computed for both these data sets separately. These statistical values are depicted for the 1000 hPa to 800 hPa layer and the 800 hPa to 400 hPa layer (Fig. 8).

The mean of the normalized differences shows different patterns for the "dry" and "wet" data sets at both Guam and Mi-namidaitojima. At Guam, we see a dry bias of 6 % to 14 % in the 1000 hPa to 800 hPa layer for all RO retrievals for the "dry" data set (Fig. 8, left). We assume that the dry air intrusions and sharp humidity transitions above the PBL with associated super-refraction conditions are primarily responsible for the negative $N$ and thus negative $q$ bias at Guam. The RO biases in the 800 hPa to 400 hPa layer vary around zero ($-4\,\%$ to $2\,\%$). For the "wet" data set, the mean RO differences from ERA vary significantly in the 800 hPa to 400 hPa from 0 % to 16 %, while the bias in the 1000 hPa to 800 hPa layer is between 0 % and $-5\,\%$ for the RO retrievals. At Minamidaitojima, the RO data sets show smaller and similar biases for both pressure layers (Fig. 8, right). The "dry" RO data set has no bias in the 800 hPa to 400 hPa layer and very small biases (2 % to 5 %) in the





1000 hPa to 800 hPa layer. The bias of the "wet" RO data set ranges from $-4\,\%$ to $4\,\%$ in the 800 hPa to 400 hPa layer and from $-4\,\%$ to $2\,\%$ in the 1000 hPa to 800 hPa layer. Overall, we conclude that there are no major differences in the RO error characteristics between the "dry" and "wet" data sets and between the two pressure layers at Minamidaitojima, in contrast to Guam where background humidity conditions clearly matter for the different error characteristics.

AIRS clearly shows a strong dry bias for both pressure layers for wet background conditions. The bias is stronger at Minamidaitojima, reaching more than $-30\,\%$ in the 800 hPa to 400 hPa layer, and $-20\,\%$ in the 1000 hPa to 800 hPa layer. For dry conditions, the AIRS bias ranges from $-8\,\%$ to $2\,\%$ for all locations and pressure layers. This agrees well with the small bias seen in the regions of dry air intrusions (December to June) in the profile time series (Fig. 4, bottom).

Finally, the RS shows a small wet bias in the 1000 hPa to 800 hPa layer for both the "dry" and "wet" data sets and both
locations. In the 800 hPa to 400 hPa layer, the "dry" data set shows a large wet bias, which is likely due to the VIZ/Sippican B2 sensor's poor performance in dry conditions (see Section 2.3). At Minamidaitojima, both the "dry" and "wet" data sets show a dry bias in the 800 hPa to 400 hPa layer, as described in Vömel et al. (2007) for the Vaisala RS92 sensor for higher altitudes due to a radiation bias.

## 5   Variability during typhoon passages

We used the subtropical RS station Ishigakijima to investigate how the different data sets perform during the extreme conditions of typhoon passages. In 2007, six typhoons passed Ishigakijima within 350 km (the tracks and other details of the typhoons can be found online[9]):

  – Typhoon #4, July 6–16, date of closest approach (320 km): July 12, as typhoon category 4
  – Typhoon #7, August 4–10, date of closest approach (260 km): August 7, as typhoon category 1
– Super Typhoon #9, August 11–19, date of closest approach (300 km): August 17, as typhoon category 4
  – Typhoon #12, September 11–17, date of closest approach (330 km): September 14, as typhoon category 4
  – Super Typhoon #13, September 14–20, date of closest approach (40 km): September 18, as typhoon category 3
  – Super Typhoon #17, October 1–8, date of closest approach (110 km): October 6, as typhoon category 4

The time series of differences to ERA $q$ for Ishigakijima do not show a specific bias during typhoon passages, which indicates
that all data sets as well as ERA report a signal similar in magnitude during the typhoon passages.

We computed the ERA average over the July to October time range (CLIMO$_{\text{JulOct}}$) to create a typhoon season climatology. We then compared all data sets to CLIMO$_{\text{JulOct}}$ to see how $q$ and $T$ deviate from the summer average during the passage of a typhoon. All data sets show a rapid increase in humidity (Fig. 9) and higher temperature values (not shown) as the typhoons approach and pass close to Ishigakijima. The signal is strongest in the layers above 600 hPa. GFS, RS, and all RO retrievals
show similar results. The AIRS moist deviation during a passage is much weaker than for any other data set, likely because of all the cloud cover associated with the typhoons, which limits the AIRS retrievals.

[9]http://weather.unisys.com/hurricane/w_pacific/2007/





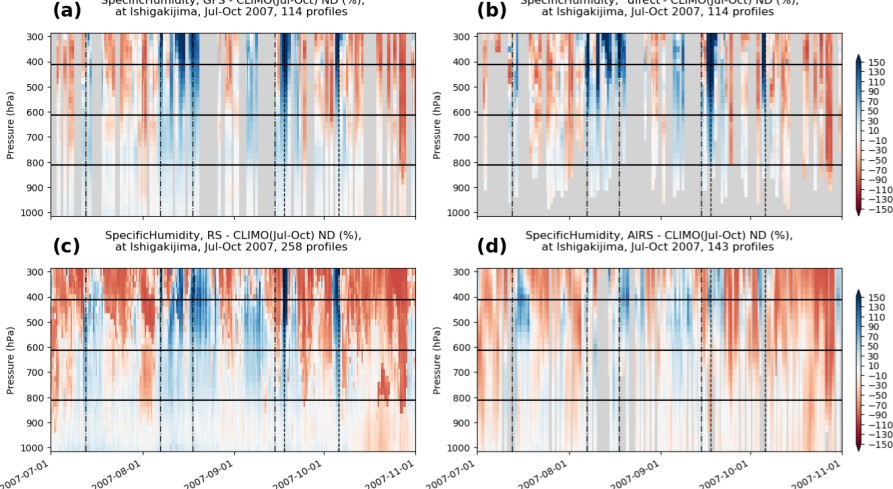

**Figure 9.** The $q$ difference from CLIMO$_{\text{JulOct}}$ for (a) GFS, (b) UCAR direct, (c) RS, and (d) AIRS shows increased humidity during typhoon passages near Ishigakijima. Typhoon passages are marked by vertical lines (dashed: closer than 110 km, dash-dotted: closer than 330 km).

All data sets show increased temperature during the typhoon event (not shown), especially in the upper troposphere. The UCAR and WEGC 1D-Var show a similar $T$ structure. The signal also agrees well with the GFS $T$ signal. Both direct retrievals (UCAR and JPL) do not provide physical temperature information for the troposphere.

The refractivity signal is the combined signal of increased $q$ (increases $N$) and increased $T$ (decreases $N$), as shown by Eq. (1). Up to about 400 hPa, the signal of increased $q$ overpowers the signal of increased $T$ in $N$ (not shown), leading to increased $N$ during the typhoon passage.

The signals in $q$, $T$, and $N$ during a typhoon passage are similar for Minamidaitojima and Naze (not shown), but fewer typhoons passed in close proximity to these two stations.

## 6   Structural uncertainty of RO

Since we have data from several RO retrievals available, we have the opportunity to compute the structural uncertainty of RO humidity for our data set. First we create sub-data sets, which are limited to the profiles and pressure levels that are available for all four RO humidity retrievals. The sub-data set for Guam consists of 141 profiles, and the sub-data set for the combined Japanese stations (since atmospheric conditions are very similar among them) consists of 543 profiles. For each retrieval, the normalized deviation for $N$ and $q$ from the inter-center mean is computed (per pressure level):

$$\Delta X = \frac{1}{k} \sum_k X_k - \overline{X}_k^{\text{inter-center}} * \frac{100}{\overline{X}_{\text{annual}}^{\text{ERA}}} \tag{4}$$

where $\Delta X$ is the deviation of a particular RO retrieval and $k$ indicates the profile number.



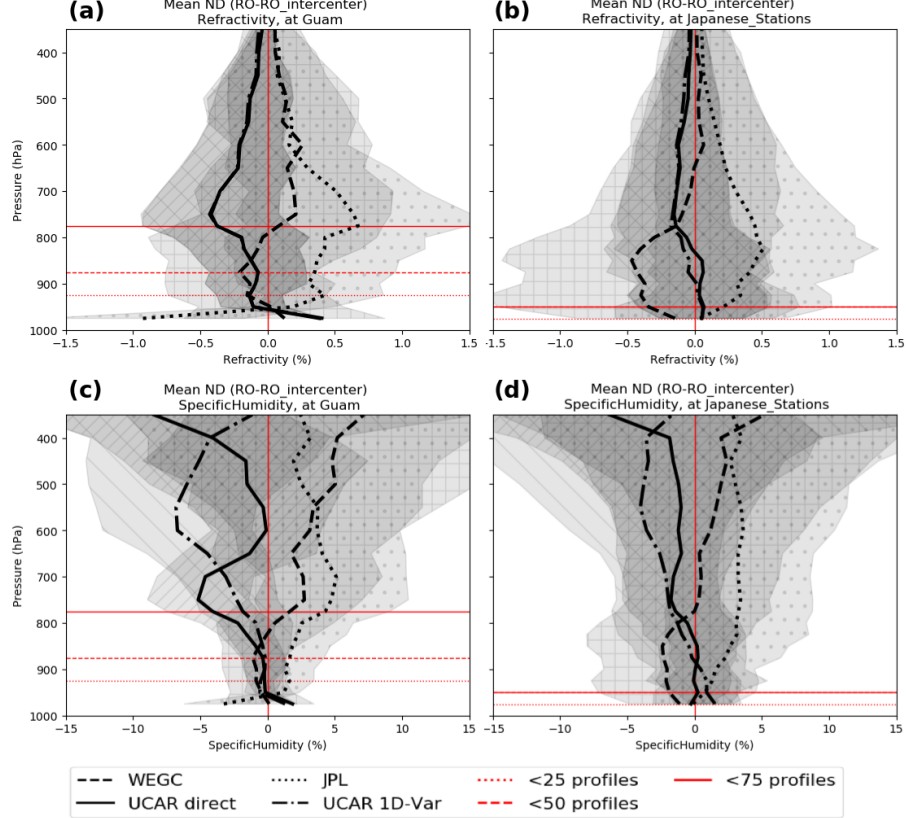

**Figure 10.** Deviations for four RO retrievals from the intercenter mean for refractivity (top) and specific humidity (bottom) at Guam (left) and the Japanese stations (right). The standard deviation of each data set is shown as shaded and hatched: UCAR direct: ////; UCAR 1D-Var: \\\; WEGC 1D-Var: +++; JPL direct: ····. Red horizontal lines indicate the number of profile pairs at that pressure level.

Figure 10 shows the mean deviations of the four RO retrievals from the inter-center mean for Guam (left) and all three Japanese stations combined (right) for $N$ (top) and $q$ (bottom). Cut-off pressure is 350 hPa since JPL does not provide humidity data above that level. For $N$ (Fig. 10 (a,b)), the mean deviation from the inter-center mean it is largest between 900 hPa and 700 hPa for all data sets (maximum: $\sim 0.7\%$)), and decreases to about 0.1 % at 350 hPa (about 8 km) at both locations. This

5   result agrees well with the estimate of Ho et al. (2009), who showed that the absolute values of fractional $N$ anomalies among four centers (UCAR, WEGC, JPL, and GFZ (German Research Centre for Geosciences)) are 0.2 % from 8 km to 25 km altitude. For $q$ (Fig. 10 (c,d)), the structural uncertainty generally increases with increasing altitude (since the impact of water vapor on $N$ decreases with increasing altitude). At Guam, it is about 2 % in the PBL, increases sharply to 5 % around 800 hPa, and stays around 5 % to 8 % above. At the Japanese stations, the structural uncertainty increases constantly with increasing altitude, from

10   2 % close to the surface to 5 % at 400 hPa. At both locations, the center anomalies increase sharply at 350 hPa, which indicates again that RO derived humidity has high uncertainty at and above that level.



## 7    Conclusions

We compared three observational data sets (radio occultation (RO), radiosondes (RS), and AIRS) and two model data sets (ERA and GFS) over the year 2007. Rather than comparing averages over larger time scales and regions, we compared individual profiles over specific locations (in the tropical and subtropical West Pacific). The data sets that were compared to ERA, which we considered the reference data set, include profiles from four different RO retrievals (UCAR direct, UCAR 1D-Var, WEGC 1D-Var, JPL direct), RS, AIRS, GFS analysis, ERA PERSIST, and ERA CLIMO (the last two to set a quality baseline). We studied both the time series of profile pairs as well as the mean and RMS computed for the four seasons and three pressure layers (1000 hPa to 800 hPa, 800 hPa to 600 hPa, and 600 hPa to 400 hPa). As expected, we found different characteristics for each data set. Our main conclusions are:

1. For all four RO humidity retrievals, the magnitude of the mean biases are smaller or comparable to those of the RS and AIRS in 800 hPa to 400 hPa layer. Above 600 hPa, differences between the various RO humidity retrievals generally become larger.

2. All data sets have smaller RMS differences than both CLIMO and PERSIST. The exception is the tropical 1000 hPa to 800 hPa layer, where all RMS values are comparable in magnitude due to the nearly constant humidity conditions throughout the year. This confirms that all observational data sets contribute valuable information compared to persistence and climatology.

3. The RMS of all RO retrievals is comparable or lower than the RMS of the RS and AIRS for all pressure layers below 400 hPa, which confirms the high quality of RO profiles. The agreement among the four different retrievals of specific humidity in the lower and middle troposphere validates the stability of the four retrievals.

4. In the time series, the four RO retrievals agree within 10 % in the 1000 hPa to 600 hPa layer. Differences become larger in the 600 hPa to 400 hPa layer, where the UCAR 1D-Var gets drier, the UCAR direct alternates between too dry and too wet, and both the WEGC 1D-Var and JPL direct become too wet. Since water vapor decreases exponentially with altitude, the retrieval becomes more and more sensitive to the prescribed temperature, which can lead to larger humidity differences.

5. The structural uncertainty of RO humidity retrievals is estimated from anomalies of RO retrievals from the inter-center mean. Maximum differences among retrievals from 1000 hPa to 400 hPa are between 1 % and 0.2 % for refractivity, and 3 % and 10 % for specific humidity.

6. RO has the potential to contribute valuable information on water vapor via data assimilation in the mid and lower troposphere, especially when high-quality RS are unavailable (southern hemisphere, over oceans). In contrast to infrared or microwave sounders, RO can resolve strong vertical gradients of humidity.

7. AIRS is biased dry throughout the entire troposphere, as noted previously (Wong et al., 2015). This bias is particularly strong for wet atmospheric conditions.



8. All data sets show increased humidity and temperature values during a typhoon passage. Differences from ERA do not change noticeably during a typhoon passage, indicating that all data sets and ERA report a signal that is similar in magnitude during the typhoon passages.

Our results support the findings of Vergados et al. (2017), e.g. the relative dryness of the UCAR 1D-Var and wetness of the
JPL RO humidity retrieval, and the dry bias of AIRS. While Vergados et al. (2017) draw their conclusions from large-scale multi-year climatologies, we use high resolution time series to depict the short-term and small scale variability of humidity, and add results below 700 hPa, where the tropospheric water vapor content is highest.

We conclude that RO humidity retrievals have comparable or better accuracy than both standard RS and AIRS data at the four tropical and subtropical locations studied here above 800 hPa, as well as below 800 hPa if super-refraction is absent. Featuring
global coverage and high vertical resolution, RO should have a large positive impact on improving the water vapor analysis in data assimilation in the lower and mid troposphere.

*Code availability.* Code will be made available by the author upon request.

*Data availability.* Data can be made available from authors upon request.

*Author contributions.* T. Rieckh and R. Anthes formulated the initial idea and developed the design of this work. W. Randel, U. Foelsche,
S.-P. Ho contributed ideas and provided valuable feedback. T. Rieckh collected the data, performed all the computational work and coding necessary, performed the analysis, and prepared the manuscript. R. Anthes contributed significantly to the data analysis and the writing.

*Competing interests.* The authors declare that they have no conflict of interest.

*Acknowledgements.* Rieckh, Anthes, and Ho were supported by the NSF-NASA grant AGS-1522830. Randel was supported through the NSF-UCAR cooperative agreement for the management of NCAR and NASA RNSS Science Team grant NNX16AK37G. The ERA data
were provided by the Data Support Section of the Computational and Information Systems Laboratory, NCAR, which is sponsored by the NSF. We thank the COSMIC CDAAC team for providing the RO Level-2 data. We also thank Drs. Eric DeWeaver (NSF) and Jack Kaye (NASA) for their long-term advice and support to the COSMIC science program. COSMIC is supported by the National Space Office (NSPO) of Taiwan, NSF, NASA, NOAA, and the U.S. Air Force. We thank the WEGC processing team members (M. Schwärz, F. Ladstädter, B. Angerer) for providing the OPSv5.6 RO data, and we thank Robert Khachikyan for making the JPL RO retrievals publicly available through
the AGAPE interactive search tool. We thank NASA Earth Observing System Data and Information System for making the AIRS data publicly available. We thank the team of the Integrated Global Radiosonde Archive for documenting and providing global radiosonde data.



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
