# Peer review of "Evaluating tropospheric humidity from GPS radio occultation, radiosonde, and AIRS from high-resolution time series"

_Atmospheric Measurement Techniques, 2017_

## Referee Comment (RC1) · Anonymous Referee #1 · 20 Feb 2018

The manuscript "Evaluating tropospheric humidity from GPS radio occultation, radiosonde, and AIRS from high-resolution time series" by Therese Rieckh et al. report validation of GPS RO with different retrieval algorithms for specific humidity against ERA-Interim. Also reported are similar validation for radiosonde measurements and AIRS V6 retrievals. The uniqueness of this article is that the authors focus on 4 locations to highlight high-resolution temporal time series and special events of typhoon passages. I think that this work provides information that the public needs to know when using GPS RO specific humidity data. However, I do have a few comments of clarification.

[Figure]

**Comments**

1. The work uses ERA-Interim as reference data to quantify "biases" of other datasets. However, comparison to AMSR total column water vapor shows that ERA-Interim may have dry bias in certain atmospheric conditions. The author also states in Pg. 3 Line 13 that previous work has shown ECMWF reanalysis being drier compared to MERRA and JPL RO (although there may be differences between ECMWF reanalysis and ERA-Interim). If the authors think ERA-Interim can be guaranteed as the "truth" at least at the 4 locations discussed in the paper, they need to clarify with reasons or references to support this. Otherwise, the authors may consider to mitigate the wording such as "biases" when referring to differences between other datasets and ERA-Interim.

2. Different datasets have different footprints as mentioned in the paper. Therefore, AIRS specific humidity represents average value within the 45-km AIRS footprint, while radiosondes are point measurement. While GPS RO is occultation, its equivalent horizontal resolution may be lower. How these differences of resolution of different datasets influence the uncertainty estimates when compared to ERA-Interim with a resolution of 0.7°?

3. As mentioned on Page 8 Lines 9-13, different datasets have different quality flags that result in different sampling sizes after paired up with ERA-Interim. As quality flag of a particular dataset favors particular atmospheric conditions (e.g., AIRS quality flags favor conditions of less deep-thick clouds). How may these differences in sample sizes influence the general conclusions of the work? For example, if one constraints all datasets to have the same samples after paired up with ERA-Interim, will this give different patterns for plots like Figs. 3-8?

4. Super-refraction seems to be a big problem for GPS retrievals. The authors may consider including some discussion of how the users of GPS can know if bad quality of retrievals caused by super-refraction exists in a particular profile. Or if GPS datasets provide quality flags to inform users if such events occur?

[Figure]

Specific Comments:

Figure 2 caption: "Scatter plots of normalized q for 7 days..." It seems that these are not "normalized q" as the units are g/kg. Normalized q should have no unit.

Pg. 11 Lines 5-8: Such dubious radiosonde profiles with constant humidity profiles are easily detected and should be excluded from the matched up pairs. Otherwise, the comparison is unfair for radiosondes. If such data are excluded, will the plots of Fig. 4a, Figs 6 and 8 be drastically changed?

Pg. 12 Lines 2-3: For deep convective clouds (thick clouds with high cloud top), the dry bias is throughout the troposphere. For low-level, thick clouds (stratocumulus or stratus), the large dry bias is confined in the lower troposphere.

Pg. 12 Lines 12-Pg 13 Lines 1-2: It is good that the authors caution the problem in ERA-Interim. But in general, can ERA-Interim be regarded as a perfect "truth"? (See Main Comment #1).

Pg. 18 Eqn. (4): Need a bracket for (Xk - Xkinter-center) in the equation before multiplied by 100/XERAannual.

Pg. 18 Line 14: Need to spend a sentence or two to explain what "inter-center mean" means. Or use an equation to tell readers how it's calculated.

Pg. 20 and Pg. 21: The authors may consider pointing at the figure number that supports each conclusive bullet.

Pg. 21 Lines 10-11: "...RO should have a large positive impact on improving the water vapor analysis in data assimilation in the lower and mid troposphere." Is this statement contradicting the claim on Pg. 4 Lines 10-14 that "RO makes a relatively small contribution in the ERA reanalysis."?

---

## Referee Comment (RC2) · Anonymous Referee #2 · 22 Feb 2018

**Date: February 22, 2018**

Manuscript #:     amt-2017-486
Manuscript title: ***Evaluating tropospheric humidity from GPS radio occultation, radiosonde, and AIRS from high-resolution time series***

**Brief Summary of the Manuscript**
This manuscript studies the statistical differences of specific humidity among numerous data sets, including GPS radio occultations from various processing centers, AIRS, ERA-Interim, and radiosondes. The analysis is performed over specific geographic locations, which are characterized by different environmental conditions. The authors use a year-worth of data, in 2007, so that they have an increased number of GPS radio occultation sampling. The advantage of this investigation over previous research is that it quantifies the statistical differences of specific humidity among all the different data sets in short temporal scales and not in an ensemble climatological mean. The manuscript contains useful results that are worth publishing in the journal of Atmospheric Measurement Techniques, only after major revisions, according to the guidelines provided below.

**Comments:**
1) **Page 3, Line 24.** The objectives are too general and slightly confusing. Please, re-write this section. For example, I am not sure what "... quantify RO humidity retrievals..." or "...quantify how these RO humidity..." mean. I recommend be specific on the objectives, as you describe them in the next paragraph.

2) **Section 2.1.** I recommend including the specific humidity accuracy for each center, including appropriate references.

3) **Section 3.1, Figure 2.** The authors should provide more discussion on the observed differences among the data sets, with respect to ERA-Interim, and try to provide an explanation of where these differences may be coming from. E.g., discuss the 1000-400 hPa dry bias seen in UCAR 1D-Var, JPL direct, and GFS. Also, at 200-100 hPa, the 1D-Var approach (UCAR and WEGC) show an excellent agreement with ERA-Interim, whereas the rest of the data sets show much larger deviations. The authors should describe that and try to explain the observed behavior. Additionally, the standard deviations and RMS of all the regressions could provide a quantitative insight on the degree of agreement with respect to ERA-Interim.

4) **Section 3.2, Page 12.** Similar to the previous comment, it would be nice if the authors could provide a more detailed discussion on the differences of each data set with respect to ERA-Interim at the GUAM location and explain these differences within the context of the environmental conditions over this region and within the context of how each profile has been retrieved by each center.

5) **Page 12, Lines 14-15.** I find this statement a bit strong, because it may exclude other possible sources that could cause the observed discrepancies. Could it also be that ERA-Interim may be overestimating the degree of entrainment above the PBL, thus introducing more water vapor in the free troposphere aloft? Have the authors considered convection as another possible source of such discrepancies? Also, it should read: "Figure 5 shows that all data sets are dry-biased with respect to..." In Figure 5, the title of the color bar is missing.

6) **Figures 6 and 7.** The title is missing from the x-axis. Also, the definition of ND assumes than ND is unitless, yet the differences here are given as percentage. Either modify the definition of the ND, or redo the figure accordingly. Perhaps, add 2-3 sentences to explain to the reader what the smaller RMS values physically mean in the radio occultations (e.g., smaller scatter, steadier daily variability, better long-term robustness, better accuracy... something along these lines, so that the reader attach a physical interpretation to the results). This way, I believe the analysis becomes clearer.

7) **Page 16, Line 2.** It has not become clear from the discussion so far how different atmospheric conditions could influence the data biases. It would be great if there were a transitioning paragraph before Section 4 that summarizes in 3 sentences the conclusions of Section 3. I believe this would be a smooth transition.

8) **Figure 8.** At 800-400 hPa over Guam, at 1000-800 hPa over Mina, and at 800-400 hPa over Mina, the three orange asterisks that indicate values about -21%, -18%, and -35%, respectively, could they be outliers? Also, it would be good to include the latitude and the longitude of each station at the beginning.

9) **Page 17, Line 5.** Could the observed AIRS dry bias be due to cloud-contaminated radiances? And therefore, the AIRS statistical differences might be statistically insignificant? I think it would be good if this were mentioned in the interpretation of the results for completeness, unless only AIRS cloud-cleared pixels are used in the analysis.

10) **Page 17, Line 29.** Any physical explanation as to why the signal is strongest in the layers above 600 hPa?

11) **Page 18, Lines 4-6.** Although this may be true, there have been no results showing refractivity variations. Therefore, aren't these lines out of context? What purpose do they serve? On another note, I would think that due to deep convection within the eye and eyewall regions of a typhoon, together with the water vapor entrainment and vertical overshooting from the well-mixed moist layer that sits at the bottom of the typhoon, there would be an increase in the water vapor concentration in the free troposphere that could lead to refractivity increase high up based on Eq. (1). Perhaps, this could be an explanation to my previous comment above?

12) **Section 6.** Why the inter-center mean and not the GUAM sub-data set? Is it because there is no "true" RO data set, and thus an inter-center mean is regarded more realistic? But then again, wouldn't the inter-center mean smooth out differences? Why not use the GUAM radiosonde data set as the "true" and redo Figure 10? Would this change the conclusions of the analysis?

13) **Page 19, Line 3.** You mean the absolute value of the mean deviation?

14) **Figure 10.** More discussion is required in the refractivity analysis. E.g., over GUAM, the JPL and WEGC refractivity differences are systematically larger than the inter-center mean above about 800 hPa and the JPL refractivity difference is larger even down to 950 hPa. Over the Japanese stations, the JPL refractivity difference is systematically larger than the inter-center mean. Any explanation as to why these may be? Could these be associated with the different environmental conditions over GUAM and the Japanese stations? Additionally, the figure caption in Figure 10 needs fixing at the part where the authors describe what line represents each data set.

15) **Section 7.** I feel that the conclusion section needs re-writing, in order to focus on the objectives of this investigation. For example, these eight concluding remarks could be summarized into one single paragraph and then a second paragraph should describe the findings of this investigation regarding the behavior of the radio occultations within the context of the: a) wet vs dry conditions, b) typhoon passages, and c) different geographic locations.

---

## Author Comment (AC1) · 29 Mar 2018

scrartcl natbib

**Authors' response to referee #1**

We thank the anonymous referee #1 for the review and comments. We will implement the following changes according to the referee's suggestions. We have answered all comments below (for easier comparison the referee comments are included in italics). All page and line number refer to the originally submitted manuscript.

[Figure]

**Comments:**

*1. The work uses ERA-Interim as reference data to quantify "biases" of other datasets. However, comparison to AMSR total column water vapor shows that ERA-Interim may have dry bias in certain atmospheric conditions. The author also states in Pg. 3 Line 13 that previous work has shown ECMWF reanalysis being drier compared to MERRA and JPL RO (although there may be differences between ECMWF reanalysis and ERA-Interim). If the authors think ERA-Interim can be guaranteed as the "truth" at least at the 4 locations discussed in the paper, they need to clarify with reasons or references to support this. Otherwise, the authors may consider to mitigate the wording such as "biases" when referring to differences between other datasets and ERA-Interim.*

The reviewer brings up a good point. We have clarified in the paper that we do not consider ERA-Interim to be the "truth", but rather use it as a common reference for comparison of all the data sets. We need a baseline for our comparison, and the ERA is the most suitable data set for that purpose. ERA assimilates a high number of quality-controlled observations in a research (rather than operational) mode, which should overall minimize variability and bias around the true values. ERA also is a complete date set (unlike the observational data sets) because it has data at all comparison points and at all times. Furthermore, even though ERA is not assumed to be an absolute "truth" and without errors, ERA has the smallest error variances when compared to RO, RS, and GFS as shown in the related paper (Anthes and Rieckh, 2018).

Thus we modified the first two sentences in the section describing ERA (Sec. 2.2, P. 6, L. 13–17) to clarify: "We use the ERA as a reference (or baseline) for our comparisons. We do not consider ERA as "truth", but we do consider the ERA to be the most accurate data set (Anthes and Rieckh, 2018), because it uses quality-checked observations with a 4D-Var data assimilation scheme and an accurate forecast model

in a research mode to produce the variables of interest here (temperature and water vapor) on a $0.7° \times 0.7°$ grid. In 2007 ERA assimilates measurements from many different observing techniques, including RS observations, AIRS radiances, and RO bending angles (Dee et al., 2011). Thus, when using the word "bias" for a data set in a comparison, we refer to the bias difference with respect to ERA."

Furthermore, we edited the following sentences: P.11 L.3, P14 L.2, P.16, L.1 (section title), P.16 L.2–3, P.17 L.1, and P.20 L.10.

*2. Different datasets have different footprints as mentioned in the paper. Therefore, AIRS specific humidity represents average value within the 45 km AIRS footprint, while radiosondes are point measurement. While GPS RO is occultation, its equivalent horizontal resolution may be lower. How these differences of resolution of different datasets influence the uncertainty estimates when compared to ERA-Interim with a resolution of 0.7° ?*

Since RS is a point measurement, it has the potential to show variability that occurs on smaller spatial scales compared to AIRS ($\sim$45 km average), ERA ($0.7°\approx 78$ km and less, depending on the latitude), and RO (average over $\sim$200 km). This makes AIRS, RO, and ERA fairly comparable in terms of horizontal resolution. To account for the larger vertical variability of RS due to its ability to detect small scale features, we tested vertically averaging the RS profiles over pressure layers before interpolating to the common 25 hPa grid. Since results were very similar to the original approach (interpolation only), we used the original approach throughout the paper.

*3. As mentioned on Page 8 Lines 9–13, different datasets have different quality flags that result in different sampling sizes after paired up with ERA-Interim. As quality flag of a particular dataset favors particular atmospheric conditions (e.g., AIRS quality flags*

*favor conditions of less deep-thick clouds). How may these differences in sample sizes influence the general conclusions of the work? For example, if one constraints all datasets to have the same samples after paired up with ERA-Interim, will this give different patterns for plots like Figs. 3–8?*

Our goal was to maximize the number of co-locations and show the results of all the data set compared to ERA. Generally, a higher number of co-locations will create a more accurate and complete picture. If we restricted all data sets to the same co-locations and sample size, we would have removed a lot of information. E.g., AIRS passes over the observed region at around 4:30–5:30 UTC and again at around 17:00–18:00 UTC, while RS are launched around noon and midnight UTC, which makes a common co-location within less than 5 hours impossible.

*4. Super-refraction seems to be a big problem for GPS retrievals. The authors may consider including some discussion of how the users of GPS can know if bad quality of retrievals caused by super-refraction exists in a particular profile. Or if GPS datasets provide quality flags to inform users if such events occur?*

Super-refraction (SR) can generally be detected on an individual profile basis by identifying profiles with a very sharp change in bending angle and a refractivity bias with respect to another, unbiased data set. This usually occurs in the tropics in the lower troposphere, and often at the top of the boundary layer. However, as far as we know, a robust method that can be applied operationally has not been developed so far. The authors are not aware that any RO processing center flags profiles that experience SR.

**Specific Comments:**

*Figure 2 caption: "Scatter plots of normalized q for 7 days..." It seems that these are not "normalized q" as the units are g/kg. Normalized q should have no unit.*

Thank you, the reviewer is correct. we removed the work "normalized" from that sentence.

*Pg. 11 Lines 5–8: Such dubious radiosonde profiles with constant humidity profiles are easily detected and should be excluded from the matched up pairs. Otherwise, the comparison is unfair for radiosondes. If such data are excluded, will the plots of Fig. 4a, Figs 6 and 8 be drastically changed?*

We do not think any portions of any of our data sets should be removed on the basis of possible or likely errors. One of the points of the comparisons is to identify such errors. Each data set may have its own set of errors or issues. To be fair, all data sets should be compared in their complete form as they passed internal (data set specific) quality control and are available for the research community. One of the purposes of our comparison study is to identify strengths and weaknesses, including errors, of all the data sets, in order that the providers of the data sets may improve their accuracy and to make users aware of the full characteristics of the data sets.

*Pg. 12 Lines 2–3: For deep convective clouds (thick clouds with high cloud top), the dry bias is throughout the troposphere. For low-level, thick clouds (stratocumulus or stratus), the large dry bias is confined in the lower troposphere.*

We reformulated the sentence on P.12 L.1–3 to: "They found reduced dry biases in the middle troposphere under thin clouds, but large dry biases in the lower troposphere (>30%) associated with low thick clouds, and dry biases throughout the troposphere in the presence of high thick clouds."

*Pg. 12 Line 12 – Pg 13 Line 2: It is good that the authors caution the problem in ERA-Interim. But in general, can ERA-Interim be regarded as a perfect "truth"? (See Main Comment #1).*

Please see our response to the reviewer's main comment #1.

*Pg. 18 Eqn. (4): Need a bracket for (Xk - Xkinter-center) in the equation before multiplied by 100/XERAannual.*

Thank you, we added parenthesis in the equation.

*Pg. 18 Line 14: Need to spend a sentence or two to explain what "inter-center mean" means. Or use an equation to tell readers how it's calculated.*

We changed the sentence on P.18 L.16 after Eq. (4) to: "where $k$ indicates the profile number, $\overline{X}_k^{\text{inter-center}}$ is the inter-center average for the $k^{\text{th}}$ profile $(1/4 \cdot (X_k^{\text{UCAR direct}} + X_k^{\text{UCAR 1D-Var}} + X_k^{\text{JPL direct}} + X_k^{\text{WEGC 1D-Var}})$, and $\Delta X$ is the deviation (of $q$ or $N$) of one particular RO retrieval from the inter-center average."

*Pg. 20 and Pg. 21: The authors may consider pointing at the figure number that supports each conclusive bullet.*

Thank you. Figure numbers in the conclusions are now added.

*Pg. 21 Lines 10–11: "...RO should have a large positive impact on improving the water vapor analysis in data assimilation in the lower and mid troposphere." Is this statement contradicting the claim on Pg. 4 Lines 10–14 that "RO makes a relatively small contribution in the ERA reanalysis."?*

In our view, the problem is that with current models, RO makes a relatively small contribution to the moisture analysis in the lower and mid troposphere because the assigned errors in the data assimilation process are too large, and so RO observations are not weighted heavily enough. To make a clearer statement, we rephrased

this sentence to: "If assigned smaller errors (and therefore greater weights) in the assimilation process, RO could have a positive impact on improving the water vapor analysis in data assimilation in the lower and mid troposphere."

**References**

Anthes, R. and Rieckh, T. (2018). Estimating observation and model error variances using multiple data sets. *Atmos. Meas. Tech. Discuss.* in review.

Dee, D. P., Uppala, S. M., Simmons, A. J., Berrisford, P., Poli, P., Kobayashi, S., Andrae, U., Balmaseda, M. A., Balsamo, G., Bauer, P., Bechtold, P., Beljaars, A. C. M., van de Berg, L., Bidlot, J., Bormann, N., Delsol, C., Dragani, R., Fuentes, M., Geer, A. J., Haimberger, L., Healy, S. B., Hersbach, H., Hólm, E. V., Isaksen, L., Kållberg, P., Köhler, M., Matricardi, M., McNally, A. P., Monge-Sanz, B. M., Morcrette, J.-J., Park, B.-K., Peubey, C., de Rosnay, P., Tavolato, C., Thépaut, J.-N., and Vitart, F. (2011). The ERA-Interim reanalysis: configuration and performance of the data assimilation system. *Quart. J. Roy. Meteor. Soc.*, 137(656):553–597.

---

## Author Comment (AC2) · 30 Mar 2018

scrartcl natbib

**Authors' response to referee #2**

We thank the anonymous referee #2 for the review and thoughtful comments and suggestions. We have responded to all comments below (for easier comparison the referee comments are included in italics). All page and line number refer to the originally submitted manuscript.

[Figure]

**Comments:**

*1) Page 3, Line 24. The objectives are too general and slightly confusing. Please, re-write this section. For example, I am not sure what "... quantify RO humidity re-trievals..." or "...quantify how these RO humidity..." mean. I recommend be specific on the objectives, as you describe them in the next paragraph.*

We believe the reviewer is referring to P.3 L.20–23. We removed the short paragraph on P.3, L.20–23 since it is considered too general and confusing, and the objectives of this study are stated more clearly and detailed in the following paragraphs. We changed the beginning sentence in the next paragraph (P.3 L.23) to make it clear that this paragraph describes the objectives:

"In this study we focus on the water vapor variability in both a temporal and spatial sense ..."

*2) Section 2.1. I recommend including the specific humidity accuracy for each center, including appropriate references.*

GPS RO accuracy for humidity is not provided by any of the data processing centers so far (WEGC is working on propagating errors throughout the retrieval to provide estimates for the final products). Some general estimates are given in literature (e.g. Ho et al. (2010); Vergados et al. (2015, 2018) and references therein), but none are for specific data processing centers. GPS RO humidity accuracy varies depending on the choice of retrieval (direct versus 1-Dimensional Variational (1D-Var) retrieval). For a direct retrieval, humidity accuracy is determined by both the quality of the a-priori temperature (Vergados et al. (2014), Fig. 1) and the refractivity accuracy. For the 1D-Var retrieval, humidity accuracy depends on the a-priori temperature and humidity

quality, the GPS RO refractivity accuracy, and the allowed error for any of the in-going parameters.

In the companion paper (Anthes and Rieckh, 2018), we present estimates of the error variances for specific humidity retrieved by the direct method and 1D-Var method at four stations in the tropics and subtropics.

We have added the following sentences on P.5 L.6: "GPS RO humidity accuracy varies depending on the choice of retrieval (direct versus 1D-Var retrieval). For a direct retrieval, humidity accuracy is determined by both the quality of the a-priori temperature (Vergados et al. (2014), Fig. 1) and the refractivity accuracy. For the 1D-Var retrieval, humidity accuracy depends on the a-priori temperature and humidity quality, the GPS RO refractivity accuracy, and the allowed error for any of the in-going parameters. A general estimate for RO $q$ accuracy is given in Vergados et al. (2018) (and references therein) as $\sim 10\,\%$–$20\,\%$."

*3) Section 3.1, Figure 2. The authors should provide more discussion on the observed differences among the data sets, with respect to ERA-Interim, and try to provide an explanation of where these differences may be coming from. E.g., discuss the 1000–400 hPa dry bias seen in UCAR 1D-Var, JPL direct, and GFS. Also, at 200–100 hPa, the 1D-Var approach (UCAR and WEGC) show an excellent agreement with ERA-Interim, whereas the rest of the data sets show much larger deviations. The authors should describe that and try to explain the observed behavior. Additionally, the standard deviations and RMS of all the regressions could provide a quantitative insight on the degree of agreement with respect to ERA-Interim.*

To address the reviewer's comment, we added the following sentences about the dry

bias seen in most RO retrievals and GFS in the 1000–400 hPa layer (P.8, L.31): "The large differences occur generally for $q$ values less than $1\,g\,kg^{-1}$, with many lower than $0.1\,g\,kg^{-1}$, which indicates dry higher altitudes (i.e. above 500 hPa). RO refractivity becomes less sensitive to water vapor at these higher altitudes and the RO retrievals of water vapor, whether direct or 1D-Var, are less reliable at these levels (Kursinski et al., 1995). The UCAR 1D-Var can also have difficulties retrieving very low humidity values (which is the case in the DJF season at Guam). If the a-priori temperature is too low, it can happen that the UCAR 1D-Var humidity values are set to zero, which would lead to a dry RO bias overall for low values of specific humidity."

To make the connection between the use of ERA/ECMWF model data as a-priori (background) information and the resultant greater agreement of the resulting RO humidity with ERA clearer, we changed the sentence on P.9 L.1–2 to: "while the UCAR 1D-Var and WEGC 1D-VAR agree very well with ERA, since they are using ERA and ECMWF short-range forecast profiles as background in the retrieval, respectively".

A quantitative insight on the degree of agreement with respect to ERA is given by the correlation coefficients (in the title of each subplot). To point this out, we added the following on P.8 L.28: "...depicting the correlation between the observational data sets and ERA at Guam (log-log correlation coefficients in the title of each panel)." We added this information in the figure caption.

We also computed the mean and standard deviation of the differences for each pressure layer and added this information in each panel of Fig. 2. Since humidity decreases exponentially with altitude, values from lower levels will have a larger influence on the result. On the other hand, the log-log axes visually emphasize small differences, e.g. the dry bias of GFS (compared to ERA) above 500 hPa in the

1000–400 hPa panel. While such differences can be important in a climate change modeling perspective, they generally not play a large role for forecasting. We added the following sentence on P.8 L.28: "Additionally, the mean and standard deviation values of the differences for each pressure layer are depicted in each panel (since values are not normalized, values from the lower levels will have a larger influence on the result)."

*4) Section 3.2, Page 12. Similar to the previous comment, it would be nice if the authors could provide a more detailed discussion on the differences of each data set with respect to ERA-Interim at the GUAM location and explain these differences within the context of the environmental conditions over this region and within the context of how each profile has been retrieved by each center.*

We provide considerable detail on each of the different data sets as well as how environmental conditions over Guam affect these differences. We tried to strike a balance between providing too little detail and providing too much. Also, we tried to avoid too much speculation about the differences in the data sets, because in many cases the reason for the differences is not known or could be due to multiple causes.

We provide details about the four RO retrievals, as well as the radiosonde, AIRS, and GFS data sets in section 2 (Data and methods). A detailed discussion of the interaction of atmospheric conditions with each data set are given on:

P.9 L.9 to P.10 L.2 about general atmospheric conditions at Guam and the impact on RO measurements ("Overall, the conditions at Guam are moist (RH$>$80 % and $q \sim 17\,\mathrm{g\,kg^{-1}}$) year-round in the boundary layer and in the mid troposphere from July to November, and dry in the mid troposphere during the rest of the year. The changing humidity pattern above 800 hPa results from the alternation of the high humidity tropical conditions and dry air intrusions from the subtropical UTLS in December

to June (Randel et al., 2016). These dry intrusions (relative humidity as low as a few percent) are very stable and suppress convection. The sharp humidity gradient between the very dry lower mid troposphere and the moist boundary layer around 800 hPa often leads to conditions of super-refraction, which results in a negative bias of $N$ and thus $q$ in the RO profiles (Xie et al., 2010).")

P.10 L.10 to P.11 L.2 on GFS ("This is essentially the layer of strong humidity variability above the bottom layer of constant (about 80 %) relative humidity. This behavior may be due to GFS difficulties in capturing the sharp transition between dry and wet conditions on the bottom of dry layers in December to June. This is supported by individual profiles (e.g. Randel et al. (2016), Fig. 4), as well as our comparison of ERA with RS (Fig. 4 (a)), which supports the ERA in this respect.")

P.11 L.5–8 on RS("At some point during the ascent, the sensor gets stuck and keeps reporting the same humidity value, which manifests itself as a dry or wet bias compared to ERA, depending on if tropospheric conditions are drier (December through May) or wetter (June through November) than the incorrect reported value.")

P.11 L.9-11 on AIRS ("The dry bias appears to be less during the dry air intrusion events in the 600 hPa to 400 hPa layer in the dry season December to June. This indicates that AIRS is less biased if the overall atmospheric conditions are dry.")

P.12 L.9–11 on RO ("...and a dry bias and/or frequent reduced penetration depth (loss of signal) in the dry season. The latter is a signature of super-refraction, which itself is caused by strong humidity gradients, usually between the planetary boundary layer and the free troposphere.")

*5) Page 12, Lines 14–15. I find this statement a bit strong, because it may exclude other possible sources that could cause the observed discrepancies. Could it also be that ERA-Interim may be overestimating the degree of entrainment above the PBL, thus introducing more water vapor in the free troposphere aloft? Have the authors considered convection as another possible source of such discrepancies? Also, it should read: "Figure 5 shows that all data sets are dry-biased with respect to...". In Figure 5, the title of the color bar is missing.*

We rewrote the paragraph on P.12 L.12 to P.13 L.2 to: "Figure 5 also shows that all RO data sets are dry-biased with respect to ERA in December through February in the 800 hPa to 600 hPa layer, which is clearly above the layer of strong humidity gradients (compare to Fig. 3 (a)). We found similar behavior in previous work. In Rieckh et al. (2017), Fig. 2, lower right panel, ERA data are given on the 775, 750, 700, and 650 hPa pressure levels (about 2.3, 2.6, 3.1, and 3.8 km). The 775 hPa and 650 hPa levels agree well with the aircraft and RO measurements; however, the two levels in between smear the sharp profile and the ERA shows humidity values $1.5\,\mathrm{g\,kg^{-1}}$ to $2.5\,\mathrm{g\,kg^{-1}}$ (20 % to 35 %) larger than the observations. Thus we conclude that the bias in Fig. 5 may not be a dry bias in RO, but could be a wet bias in ERA in the layer just above the strong humidity transition from wet (PBL) to dry (above). The assumed errors for assimilating RO in ERA are large in the lower troposphere, and all assimilated nadir viewing instruments only provide vertical resolutions of about 2 km to 3 km. Unless a nearby, approved RS contributes information locally, ERA does not have any vertically well resolved humidity data that will cause the ERA analysis to develop such sharp humidity gradients."

We have added to the Fig. 5 caption: "The color bar on the right indicates specific humidity normalized differences in %."

*6) Figures 6 and 7. The title is missing from the x-axis. Also, the definition of ND assumes than ND is unitless, yet the differences here are given as percentage. Either modify the definition of the ND, or redo the figure accordingly. Perhaps, add 2-3 sentences to explain to the reader what the smaller RMS values physically mean in the radio occultations (e.g., smaller scatter, steadier daily variability, better long-term robustness, better accuracy... something along these lines, so that the reader attach a physical interpretation to the results). This way, I believe the analysis becomes clearer.*

We added a label to the x-axis of both plots. We changed the definition of "normalized difference" in Sec. 2.4, P.8, L.21 to "$\mathrm{ND} = 100 \cdot (\mathrm{data\ set} - \mathrm{ERA})/\mathrm{CLIMO}$ (expressed as %).".

We added the following sentences before the paragraph starting on P.14, L.14: "Since the mean of the paired normalized differences is no indicator of their variability, we also show the RMS (Fig. 6, bottom). The magnitude of the RMS is a measure of the accuracy and scatter of the data compared to the reference. All data sets have . . ."

*7) Page 16, Line 2. It has not become clear from the discussion so far how different atmospheric conditions could influence the data biases. It would be great if there were a transitioning paragraph before Section 4 that summarizes in 3 sentences the conclusions of Section 3. I believe this would be a smooth transition.*

We state in Sec. 3.2 how atmospheric conditions possibly influence data biases, e.g. on:

P.9 L.13 to P.10 L.2 on RO ("The sharp humidity gradient between the very dry lower mid troposphere and the moist boundary layer around 800 hPa often leads to conditions of super-refraction, which results in a negative bias of $N$ and thus $q$ in the RO profiles (Xie et al., 2010).")

P.11 L.9 on AIRS ("The dry bias appears to be less during the dry air intrusion events in the 600 hPa to 400 hPa layer in the dry season December to June. This indicates that AIRS is less biased if the overall atmospheric conditions are dry.")

P.12 L.9–11 on RO ("...a dry bias and/or frequent reduced penetration depth (loss of signal) in the dry season. The latter is a signature of super-refraction, which itself is caused by strong humidity gradients, usually between the planetary boundary layer and the free troposphere.").

The first sentence in Sec. 4 summarizes the main findings of Sec. 3, and this is intended to serve as a transition between Secs. 3 and 4.

*8) Figure 8. At 800–400 hPa over Guam, at 1000–800 hPa over Mina, and at 800–400 hPa over Mina, the three orange asterisks that indicate values about -21%, -18%, and -35%, respectively, could they be outliers? Also, it would be good to include the latitude and the longitude of each station at the beginning.*

Figure 8 depicts mean differences of RS, RO (four retrievals), and AIRS compared to ERA. Data sets are divided into "dry" and "wet" atmospheric conditions depending on the ERA relative humidity average over the 800–400 hPa layer. We do not consider the AIRS results as outliers since they show the average difference to ERA of the "dry"/"wet" data set over an entire year.

The latitude and longitude for each stations are given in Sec. 2.3, first sentence.

*9) Page 17, Line 5. Could the observed AIRS dry bias be due to cloud-contaminated*

*radiances? And therefore, the AIRS statistical differences might be statistically insignif-icant? I think it would be good if this were mentioned in the interpretation of the results for completeness, unless only AIRS cloud-cleared pixels are used in the analysis.*

We added the following sentences in the section describing AIRS (Sec. 2.3, P.7, L.11): "The AIRS retrieval is a cloud-clearing retrieval. Susskind et al. (2003) describes the cloud-clearing process that yields the "clear" radiances from which all parameters except clouds are retrieved (Kahn et al., 2014). The humidity retrieval of Version 6 is basically the same as in Version 5, but yields improved humidity results due to the improved first guess provided by the Neural-Net start-up system, improvements in the determination of other atmospheric variables, and improvements in cloud-cleared radiances (Susskind et al., 2014)."

*10) Page 17, Line 29. Any physical explanation as to why the signal is strongest in the layers above 600 hPa?*

We modified the sentence on P.17 L.29 to: "The signal is strongest above 600 hPa, where deep convection associated with the typhoons transports large amounts of water vapor and releases latent heat in the middle and upper troposphere (Emanuel, 1991)."

*11) Page 18, Lines 4–6. Although this may be true, there have been no results showing refractivity variations. Therefore, aren't these lines out of context? What purpose do they serve? On another note, I would think that due to deep convection within the eye and eyewall regions of a typhoon, together with the water vapor entrainment and verti-cal overshooting from the well-mixed moist layer that sits at the bottom of the typhoon, there would be an increase in the water vapor concentration in the free troposphere that could lead to refractivity increase high up based on Eq. (1). Perhaps, this could be an explanation to my previous comment above?*

The authors agree with the reviewer, since $N$ is not shown, lines 4–6 are out of context and have been removed.

*12) Section 6. Why the inter-center mean and not the GUAM sub-data set? Is it because there is no "true" RO data set, and thus an inter-center mean is regarded more realistic? But then again, wouldn't the inter-center mean smooth out differences? Why not use the GUAM radiosonde data set as the "true" and redo Figure 10? Would this change the conclusions of the analysis?*

This section is focused only on the RO retrievals. To determine the structural uncertainty of RO observations, the inter-center average is commonly used as a baseline (see e.g., Steiner et al. (2013); Ho et al. (2009, 2012)). The structural uncertainty is computed to get an estimate of the variability among the various RO retrievals. We have added this information at the beginning of Sec. 6:

"Since we have data from several RO retrievals available, we have the opportunity to compute the structural uncertainty of RO humidity for our data set, following the methods of Steiner et al. (2013) and Ho et al. (2009, 2012). The structural uncertainty is computed to get an estimate of the variability among the various RO retrievals."

*13) Page 19, Line 3. You mean the absolute value of the mean deviation?*

Thank you, the reviewer is correct. We rephrased the sentence to: "For $N$ (Fig. 10 (a,b)), the absolute value of the mean deviation from the inter-center mean is largest . . . "

*14) Figure 10. More discussion is required in the refractivity analysis. E.g., over GUAM, the JPL and WEGC refractivity differences are systematically larger than the inter-center mean above about 800 hPa and the JPL refractivity difference is larger even*

*down to 950 hPa. Over the Japanese stations, the JPL refractivity difference is sys-
tematically larger than the inter-center mean. Any explanation as to why these may
be? Could these be associated with the different environmental conditions over GUAM
and the Japanese stations? Additionally, the figure caption in Figure 10 needs fixing at
the part where the authors describe what line represents each data set.*

We included the line style description into the figure caption.

Furthermore, we modified the sentences on P.19 L.3–6 to: "For $N$ (Fig. 10 (a,b)), the
absolute value of the mean deviation from the inter-center mean is largest between
900 hPa and 700 hPa for all data sets (maximum of 0.7 %), and decreases to about
0.1 % at 350 hPa (about 8 km) at both locations. The latter result agrees well with
the estimate of Ho et al. (2009), who showed that the absolute values of fractional
$N$ anomalies among four centers (UCAR, WEGC, JPL, and GFZ (German Research
Centre for Geosciences)) are 0.2 % from 8 km to 25 km altitude. The larger differences
between the various RO processing centers at lower altitudes primarily come from
different handling of profiles experiencing 1) atmospheric multipath, 2) receiver tacking
errors, and 3) super-refraction (see Ho et al. (2009) for details on the RO processing
center procedures). This is especially true for direct retrievals (such as the UCAR
direct and JPL direct), where both RO $N$ and a-priori $T$ are assigned zero error,
and the differences in Fig. 10 (a) and (b) are dominated by the previously mentioned
conditions. For 1D-Var retrievals, another potential source of differences is the $N$
error model in the respective 1D-Var retrieval. All these factors vary with latitude and
general atmospheric conditions."

*15) Section 7. I feel that the conclusion section needs re-writing, in order to focus on the
objectives of this investigation. For example, these eight concluding remarks could be
summarized into one single paragraph and then a second paragraph should describe
the findings of this investigation regarding the behavior of the radio occultations within*

*the context of the: a) wet vs dry conditions, b) typhoon passages, and c) different geographic locations.*

We thank the reviewer for this suggestion, but we think it is important to enumerate the different and disparate conclusions in crisp, short sentences as we have done rather than try to combine them into one paragraph. We have added a summary paragraph on the main results regarding the behavior of the different data sets (not just radio occultation) in the different seasons, locations, and under typhoon conditions.

We added two summary paragraphs after bullet 8 on P. 21: "We find that the alternating wet and dry seasons at Guam, together with the very sharp transition at the top of the planetary boundary layer in the dry season at Guam, are especially challenging for the RO, RS and, AIRS observational systems compared to the conditions at the subtropical Japanese locations. The results comparing the different data sets to the ERA are similar at the three Japanese RS stations.

All the observational data sets at the Japanese stations show a response to the rapid increase of water vapor throughout the troposphere during the passage of typhoons; however, the AIRS response is weaker than the RS and RO responses, probably because of the extensive clouds associated with the typhoons."

**References**

[revised manuscript text omitted]

---

## Author Response (AR2)

**Authors' response to referee #1**

We thank the anonymous referee #1 for the comment. We will implement the following changes according to the referee's suggestions (the referee comment is included in italics):

*The authors address most of my questions except for major comment 2. As indicated from the responses, the range of spatial scales for these different instruments is ∼45 km to ∼200 km, the latter is even bigger than a GCM grid cell. For temperature, it might be OK due to its relatively homogeneous structure (particularly in the tropics). For specific humidity, its spatial variability is much larger, and scales do matter. And this range of scales is not "fairly comparable" as stated in the response.*
*However, the authors can simply state the fact that the bias caused by scale differences may be minimized after averaging a large sample of data if the relative collocations of these data are random. Say, sometimes instrument 1 is located in the east of instrument 2, sometimes west, sometimes south…etc among all the samples. However, the RMS errors caused by this spatial scale differences remain. The authors should at least acknowledge this in the Method or Discussion sections. After this minor change, I think the article is acceptable.*

To address the reviewer's comment, we have added the following paragraph on page 8, line 25:

[revised manuscript text omitted]